



# Influence of GEOTRACES data distribution and misfit function choice on objective parameter retrieval in a marine zinc cycle model

Claudia Eisenring[1], Sophy E. Oliver[2,3], Samar Khatiwala[2], Gregory F. de Souza[1]

[1]Institute of Geochemistry and Petrology, ETH Zurich, Clausiusstrasse 25, Zurich, 8092, Switzerland
[2]Department of Earth Sciences, University of Oxford, South Parks Road, Oxford, OX1 3AN, UK
[3]National Oceanography Centre, Southampton, SO14 3ZH, UK

*Correspondence to*: Claudia Eisenring (claudia.eiesenring@erdw.ethz.ch)

**Abstract.** Biogeochemical model behaviour for micronutrients is typically hard to constrain because of the sparsity of observational data, the difficulty of determining parameters in situ, and uncertainties in observations and models. Here, we

assess the influence of data distribution, model uncertainty and misfit function on objective parameter optimisation in a model of the oceanic cycle of zinc (Zn), an essential micronutrient for marine phytoplankton with a long whole-ocean residence time. We aim to investigate whether observational constraints are sufficient for reconstruction of biogeochemical model behaviour, given that the Zn data coverage provided by the GEOTRACES Intermediate Data Product 2017 is sparse. Furthermore, we aim to assess how optimisation results are affected by the choice of misfit function and by confounding factors such as

analytical uncertainty in the data or biases in the model related to either seasonal variability or the larger-scale circulation. The model framework applied herein combines a marine Zn cycling model with a state-of-the-art estimation of distribution algorithm (Covariance Matrix Adaption Evolution Strategy, CMA-ES) to optimise the model towards synthetic data in an ensemble of 26 optimisations. Provided with a target field that can be perfectly reproduced by the model, optimisation results in perfect parameter retrieval regardless of data coverage. As differences between the model and the system underlying the

target field increase, the choice of misfit function can greatly impact optimisation results, while limitation of data coverage is in most cases of subordinate significance. In cases where relatively distinct model behaviours are determined with full and limited data coverage, we find considerable alignments between the two when applying a misfit metric that compensates for differences in data coverage between ocean basins.

## 1 Introduction

Marine phytoplankton account for almost 50% of global primary production (Field et al., 1998). Phytoplankton growth requires a variety of micronutrients, particularly first-row transition metals (Morel et al., 2014). In this study, we focus on zinc (Zn), which is physiologically important as a co-factor in essential enzymes (e.g. Morel and Price, 2003; Shaked et al., 2006; Morel et al., 2014), and has a high cellular abundance in phytoplankton (Moore et al., 2013; Twining and Baines, 2013).

Though Zn, like phosphorus (P), is associated with organic matter rather than with siliceous frustules (Ellwood and Hunter,

2000; Twining et al., 2003; Twining and Baines, 2013), the global distribution of dissolved Zn correlates with dissolved silicon



(Si) rather than with phosphate (PO₄; Bruland, 1980). Vance et al. (2017) revealed the key role of enhanced Zn:P uptake by diatoms in the Southern Ocean, which, when coupled to the three-dimensional ocean circulation, causes the correlation between Zn and Si on a near-global scale. Deviations from this large-scale pattern have been observed in both the Atlantic (Conway and John, 2014; Lemaitre et al., 2020) and the Pacific Ocean (Janssen and Cullen, 2015; Vance et al., 2019), and thus while the first-order significance of enhanced Southern Ocean Zn:P uptake is uncontested (Ellwood, 2008; Roshan et al., 2018; Weber et al., 2018; Middag et al., 2019), various processes have been suggested to be important distal from the Southern Ocean. Proposed water-column cycling processes include reversible scavenging (John and Conway, 2014; Weber et al., 2018), local overprinting signals related to biology (Middag et al., 2019; Vance et al., 2019), and authigenic sulphide precipitation in low-oxygen zones (Janssen and Cullen, 2015; but cf. Vance et al., 2019). Hypothesised inputs comprise sediment fluxes and atmospheric aerosol deposition in both the Atlantic (Conway and John, 2014; Lemaitre et al., 2020) and the Pacific (Conway and John, 2015; Liao et al., 2020), as well as hydrothermal input (Conway and John, 2014; Roshan et al., 2016; Lemaitre et al., 2020).

The emergence of these hypotheses is a consequence of the GEOTRACES programme, which has increased the volume of marine trace metal abundance data by orders of magnitude (Conway et al., 2021). This increase in data availability has catalysed modelling studies of marine micronutrient cycles. The first global biogeochemical models have emerged for Zn (Vance et al., 2017; de Souza et al., 2018; Roshan et al., 2018; Weber et al., 2018) as well as for a range of other bioactive metals (e.g. van Hulten et al., 2017; Richon and Tagliabue, 2019), and the understanding of the oceanic iron cycle has evolved remarkably (e.g., Tagliabue et al., 2017; Tagliabue et al., 2019; Roshan et al., 2020).

However, compared with the data for macronutrients, metal micronutrient observations remain sparsely distributed, posing one of the major difficulties faced when constraining biogeochemical models. Additionally, measurements of Zn uptake rates and cellular quotas (Zn:P) are scarce (Sunda and Huntsman, 1992), posing a challenge to modelling studies in which simulated Zn uptake must represent a variety of oceanic phytoplankton species. Further difficulties arise from commonly made assumptions regarding the precision of observations and the accuracy of the model. The combination of (i) analytical uncertainty, (ii) unresolved seasonal variability, and (iii) errors due to systematic bias in the circulation model results in a high level of parametric uncertainty, which is ideally addressed by determining model parameters through objective parameter estimation that minimises the misfit between model output and observations.

Developing a truly data-constrained view of the oceanic Zn cycle thus requires a framework that enables quantitative assessment of the explanatory skill of the above-mentioned hypotheses regarding the marine Zn cycle. To this end, we aim to assess the strengths and sensitivities of an evolutionary algorithm for constraining biogeochemical model behaviour with data, particularly given the data coverage of the GEOTRACES Intermediate Data Product 2017 version 2 (IDP2017; Schlitzer et al., 2018; Fig. 1). We do so by optimising a global ocean Zn model using a state-of-the-art optimisation algorithm, Covariance Matrix Adaptation-Evolution Strategy (CMA-ES; Hansen and Ostermeier, 2001; Hansen, 2006).

Using synthetic data that allow us full control over the "observations", we perform a suite of 26 model optimisations in order to separately assess the impact of the above-mentioned uncertainties and biases, to investigate how optimisation results are





impacted by the relatively sparse data coverage for Zn in the IDP2017, and by the choice of misfit function. Our results suggest
that optimisation with the data coverage for Zn from IDP2017 can lead to very similar biogeochemical model behaviour as
when the algorithm is provided with perfect data coverage. However, with increasing uncertainty, the results become strongly
dependent on the choice of misfit function. In such cases, misfit functions that implicitly compensate for the uneven
geographical distribution of observations in the data tend to produce more accurate model behaviour, while those that favour

the deep ocean are sensitive to systematic biases in the deep-ocean ventilation timescale of the underlying circulation model.

## 2 Methods: models and optimisation ensemble

We use a model framework that combines an offline approach for physical transport of dissolved Zn with an Estimation of
Distribution Algorithm (EDA) for optimisation of four model parameters affecting the biogeochemical cycling of Zn.
Simulations were carried out on the high-performance computing cluster *Euler* at ETH Zurich and the Cray XC40 *Piz Daint*

at the Swiss National Supercomputing Centre (CSCS). Our ensemble of 26 optimisations comprises 10 optimisation
experiment types (Sect. 2.3), each carried out with a subset of six misfit functions (Sect. 2.3.2).

### 2.1 Biogeochemical ocean model

### 2.1.1 Circulation framework

Coupled physical-biogeochemical ocean models typically have a long equilibration time due to the timescales associated with

global ocean circulation (Khatiwala, 2008; Wunsch and Heimbach, 2008). To efficiently simulate passive tracer transport, we
use the transport matrix method (TMM), which calculates the transport of dissolved species as a sequence of sparse matrix-
vector products (Khatiwala et al., 2005; Khatiwala, 2007). For our optimisation experiments, we use annual-mean transport
matrices (TMs) derived from MITgcm-2.8, a 2.8° global configuration of the MITgcm ocean general circulation model with
15 vertical levels (Marshall et al., 1997; Dutkiewicz et al., 2005). This coarse-resolution model, which is forced by

climatological winds, heat and freshwater fluxes, allows us to carry out a broad suite of optimisation experiments. In order to
approach steady state in the global Zn field, each coupled physical-biogeochemical simulation was integrated forward in time
using the TMM software (Khatiwala, 2018) for 3000 model years, using a time step of 12h for both tracer transport and
biogeochemical interactions.

### 2.1.2 Biogeochemical model of zinc

The biogeochemical Zn model used in this study is described in detail by de Souza et al. (2018). Briefly, the biological Zn
uptake term, $J_{Zn}^{uptake}$, is directly tied to that of PO$_4$ by the stoichiometric parameter $r_{Zn:P}$:

$$J_{Zn}^{uptake} = r_{Zn:P} \cdot J_{PO_4}^{uptake} \qquad (1)$$





Biological uptake of PO$_4$, $J_{PO_4}^{uptake}$, is diagnosed by a biogeochemical P cycling model based on that described in Najjar et al. (2007), in which the uptake of PO$_4$ in the surface ocean is driven by restoring surface PO$_4$ concentrations towards annually

averaged observations from World Ocean Atlas 2018 (WOA2018; Garcia et al., 2019) with a restoring timescale of 36 days. The stoichiometric parameter $r_{Zn:P}$ (mol/mol) is a nonlinear function of the concentration of free Zn (Zn$^{2+}$), arising from the phytoplankton culturing experiments of Sunda and Huntsman (1992):

$$r_{Zn:P} = \frac{a_{Zn} \cdot Zn^{2+}}{b_{Zn} + Zn^{2+}} + c_{Zn} \cdot Zn^{2+} \qquad (2)$$

Concentrations of Zn$^{2+}$ are calculated from total dissolved Zn (the tracer carried in the model) by assuming rapid equilibration

with an organic ligand with conditional stability constant $K'_{ZnL}$=10$^{10}$ M$^{-1}$ and spatially constant ligand concentration, following the approach of Ellwood and van den Berg (2000). The Zn uptake term is restricted to the euphotic zone, which comprises the uppermost two levels of MITgcm-2.8 (0–120 m). This uptake is exported downwards, where it is regenerated from an implicit particulate flux that attenuates with a power-law depth-dependence, i.e. a "Martin curve" with exponent -0.858 (Martin et al., 1987), identical to that used for P (Twining et al., 2014). All simulations are initialised with a constant Zn field corresponding

to a global ocean mean concentration of 5.4 nM (Chester and Jickells, 2012).

In our optimisation experiments, we estimate the values of parameters $a_{Zn}$, $b_{Zn}$, and $c_{Zn}$ in Eq. (2), which control different aspects of the dependency of $r_{Zn:P}$. We also optimise the organic ligand concentration $L$, which determines the concentration of Zn$^{2+}$, and thus the dependency of $r_{Zn:P}$ on total dissolved Zn. The influence of changes in each of these parameters on $r_{Zn:P}$ is illustrated in Fig. 2. All parameters optimised are assumed to be globally and temporally constant. Parameter boundaries for

optimisation were chosen relatively conservatively (i.e. broadly), since values are poorly constrained for both the ligand concentration (Bruland, 1989; Donat and Bruland, 1990; Ellwood and van den Berg, 2000; Lohan et al., 2005; Baars and Croot, 2011; Kim et al., 2015; Sinoir et al., 2016) and the parameters governing the uptake curve (Sunda and Huntsman, 1992). For the parameters $a_{Zn}$, $b_{Zn}$, and $c_{Zn}$, the lower and upper boundary is determined by subtracting 50 % from, or adding 50 % to, their minimum and maximum values reported by Sunda and Hunstman (1992). The parameter boundaries for $L$ were likewise

determined based on the range of observed values reported in Ellwood and van den Berg (2000). Parameter boundaries and reference values of each parameter are summarised in Table 1.

## 2.2 Optimisation algorithm

For parameter optimisation, our model framework relies on the Covariance Matrix Adaptation-Evolution Strategy (CMA-ES), more precisely the $\left(\frac{\mu}{\mu_w}, \lambda\right)$-CMA-ES of Hansen (2016), an Estimation of Distribution Algorithm that performs particularly

well on multi-modal functions (Hansen et al., 2010). While learning the covariance matrix in CMA-ES is analogous to learning the inverse Hessian matrix in a "classical" quasi-Newton method, CMA-ES outperforms the latter if the search landscape is non-convex or rugged (Hansen, 2016).





The CMA-ES algorithm provides a method for updating the mean and the covariance matrix of a multivariate normal search distribution, with dimensions corresponding to the number of parameters being optimised. In contrast to "conventional" evolutionary algorithms, CMA-ES updates the mean and the covariance matrix by maximising the likelihood of previously successful candidate solutions and search steps respectively (Hansen, 2006). It thus efficiently incorporates information from the entire population, while also exploiting information between generations. The latter characteristic is particularly important here, as we use a small population size with ten individuals ($\lambda$=10), so that step-size control is key in preventing the population from premature convergence. CMA-ES has been shown to be a reliable and highly competitive evolutionary algorithm for both local (Hansen and Ostermeier, 2001) and global optimisation (Hansen and Kern, 2004; Hansen, 2009). It has been tested on real-world problems including parameter calibration in a biogeochemical ocean model by Kriest et al. (2017), whose implementation of CMA-ES in C++ we employ here, via the OptClimSO package (https://doi.org/10.5281/zenodo.5517610; Tett et al., 2013; Oliver et al., 2021). In order to apply CMA-ES to a constrained problem, we use the boundary handling described in Hansen et al. (2009), in which boundaries are imposed by adding a penalty function to the calculated misfit when a parameter's distribution mean is out of bounds. As in Kriest et al. (2017), optimisation is terminated if the relative deviation of the misfits of seven individuals in a generation is smaller than $10^{-5}$, or if a predefined maximum of 200 iterations is reached. We also terminate optimisation when an individual produces a numerically very small misfit, smaller than that equivalent to a relative residual of $10^{-6}$ in each model cell.

## 2.3 Experiment setups

In all experiments, the Zn-cycling model was optimised toward synthetic observations (*target fields*) obtained from a previous model run. While we always apply the same model setup to carry out optimisation, the experiment types differ with respect to their target field. In the simplest test case, the target field is created with a physical and biogeochemical model setup identical to that being optimised. We refer to this test case as a TWIN experiment, since the target field can be perfectly reproduced by the model. The parameter values and the resulting target field of the TWIN experiment are hereafter referred to as *reference parameters* (Table 1) and *reference field* respectively. We refer to our second type of experiment as synObs (for "synthetic observations"). Target fields for the synObs experiments were created using the same reference parameter values, but either different model setups were applied to produce the target field, or it was modified a posteriori, as detailed below.

Figure 1a arranges our optimisation experiments in a conceptual raster of the degree of data limitation versus degree of complexity of uncertainties. The simplest case is given by experiment **TWIN_ALL**, in which the model is optimised towards the entire Zn field produced by a previous simulation with the same model, i.e. the reference field. Experiment **TWIN_IDP** uses this same Zn field, but limits the data available for model optimisation by subsampling it only at those spatial locations where actual Zn observations are available in the IDP2017 (Schlitzer et al., 2018; Fig. 1b, c). The subscript "IDP+" refers to those experiments in which data coverage was expanded to include the locations of high-latitude observations published more recently than the IDP2017 (Section 2.3.3; Fig. 1b, c). The remaining experiments extend our optimisation array along the axis of increasing complexity of uncertainty, as detailed below.



### 2.3.1 Types of uncertainty in synObs experiments

In order to assess the effect of various kinds of uncertainty on optimisation with full data coverage (**synObs_ALL**), and the extent to which optimisation results are affected when we additionally account for the real, imperfect data coverage of the IDP2017 (**synObs_IDP**), we separately consider three sources of uncertainty:

(i)        analytical errors in the "observations" (synObs_*[ALL/IDP]*_noise, summarised as synObs_noise),

    (ii)       lack of seasonal variability in the model (synObs_*[ALL/IDP]*_seas, summarised as synObs_seas)

    (iii)     systematic biases in the physical ocean model (synObs_*[ALL/IDP]*_circ, summarised as synObs_circ).

In order to assess the effect of observational analytical uncertainty on the optimisation results, the target fields in **synObs_noise** were obtained by perturbing the reference field with normally distributed random noise having zero mean and variance $\varepsilon$:

$\varepsilon = (0.0719 \cdot [Zn]^{0.7269})^2$                                               (3)

which is an empirical estimate of the variance of Zn concentration analyses from GEOTRACES Zn intercomparison statistics (Bruland, 2013), and assumes that the analytical errors are laboratory-independent. Any negative concentrations resulting from this perturbation were set to zero.

To investigate the influence of the lack of seasonal variability in our Zn-cycling model, **synObs_seas** experiments comprise

optimisation towards an annual-mean target field produced by a simulation with the same physical model, but with a seasonal cycle in both physical transport and biogeochemistry (Khatiwala, 2007). Our last set of experiments, **synObs_circ**, assesses the sensitivity of the optimisation to systematic biases in the circulation of the physical model. In these experiments, the target field was produced with MITgcm-ECCO, a higher-resolution version of MITgcm from the Estimating the Circulation and Climate of the Ocean (ECCO) project (Stammer et al., 2004), i.e. a different physical model than that used during optimisation

(MITgcm-2.8). In ECCO, an adjoint approach was used to adjust heat, momentum and freshwater fluxes so as to minimise the misfit between the model and a suite of observations (Wunsch and Heimbach, 2007). Climatological monthly mean transport matrices covering the 1992-2004 estimation period were extracted by Khatiwala (2007) and are annually averaged for use here. Our synObs_circ experiments aim to assess the effect of a reduction in data coverage on the optimisation results in the presence of systematic bias in the OGCM, rather than the effect of the OGCM itself, which is known to be large (Doney, 1999;

Doney et al., 2004; Najjar et al., 2007; Sinha et al., 2010; Dietze and Löptien, 2013; Löptien and Dietze, 2019; Kriest et al., 2020).

Metrics summarising similarities between the target fields of the synObs experiments and the target fields of TWIN experiments (i.e. the (reduced) reference field) are illustrated in Taylor diagrams (Taylor, 2001; Fig. 1d, e). This comparison shows that the synObs_circ target field, obtained with a different circulation model, is most distinct from the reference field

with respect to all metrics illustrated. With regard to integrated Zn export flux, the impact of simulating seasonal variability is higher: while the simulation with MITgcm-ECCO produces an export flux that is 7 % higher than the reference simulation, the seasonal MITgcm-2.8 simulation has an export flux 9 % lower. Spatial differences between target fields, and the associated export fluxes, are visualised in Fig. S1.





### 2.3.2 Misfit functions

The difference between data and model is referred to as misfit (Lynch et al., 2009), which in this study is equivalent to the model error, since the data error of synthetic observations is zero. We calculate misfit at the location of our synthetic observations. Thus, for experiments with reduced data coverage (IDP and IDP+), the model output is interpolated to the target grid before calculating misfit. In this study, we assess the applicability of six misfit metrics, which can be described using one of the following equations:

$$M = \sum_{j=1}^{N_{reg}} \sqrt{\sum_{i=1}^{N_{obs,j}} \frac{\left(m_{i,j}-o_{i,j}\right)^2}{N_{obs,j}} \cdot w_{i,j}} \tag{4}$$

$$M = \sum_{j=1}^{N_{reg}} \sqrt{\sum_{i=1}^{N_{obs,j}} \frac{\left|m_{i,j}-o_{i,j}\right|}{N_{obs,j}} \cdot w_{i,j}} \tag{5}$$

where $N_{reg}$ is the number of regions, $N_{obs,j}$ the number of observations in region $j$, and $m_{i,j}$ and $o_{i,j}$ are the modelled and the "observed" (i.e. target) Zn concentrations, respectively, at each observational point $i, j$. The local model-observation difference $m_{i,j} - o_{i,j}$ is referred to as the residual, and its squared (Eq. 4) or absolute (Eq. 5) value is weighted by $w_{i,j}$. We use four misfit

metrics based on squared residuals (Eq. 4) and two based on absolute residuals (Eq. 5). An overview of the misfit metric applied in each of our experiment types is provided in Table S1.

The four misfit metrics using squared residuals are (i) root-mean-square error (RMSE), (ii) volume-weighted RMSE (VolRMSE), (iii) variance-weighted RMSE (VarRMSE), and (iv) sum of regional RMSEs (BasinRMSE). For (i) to (iii), $N_{reg}$ equals one. In the case of RMSE, the weighting factor $w_{i,j}$ is unity for each squared residual. For VolRMSE, squared residuals

are weighted by the fractional volume of the corresponding model cell. As the vertical grid spacing of MITgcm-2.8 increases with depth and model cell volume decreases towards the poles, VolRMSE weights the deep and low-latitude ocean more strongly. Volume-weighting is frequently applied in ocean modelling studies when constraining towards observations of dissolved quantities (e.g. Kriest et al., 2017; Kwon et al., 2022).

Misfit function VarRMSE is only applied our synObs_noise experiment, in which synthetic observations were perturbed with

heteroscedastic noise. In VarRMSE, $w_{i,j}$ equals the reciprocal of the variance of the synthetic observations, $\varepsilon$ (Eq. 3). Weighting the squared residuals by $\varepsilon^{-1}$ is identical to the chi-squared statistic (e.g. Bevington and Robinson, 2003) and is frequently applied for multivariate comparison of predictions and observations, where the covariance between observational errors is assumed to be zero (e.g. Stow et al., 2009). In our calculation of variance-derived weights, residuals are calculated for synthetic observations larger than zero only.

Misfit metric BasinRMSE, which sums regional RMSEs, was only applied in experiments with reduced data coverage. Distinction between ocean regions is frequently applied when optimisations are carried out on irregularly and sparsely sampled trace metal data (e.g. Frants et al., 2016; Weber et al., 2018). For BasinRMSE, we distinguish between five ocean basins



($N_{reg} = 5$): Atlantic, Pacific, Indian Ocean, and two latitudinal sections of the Southern Ocean (40–50° S and >50° S). The resulting misfit corresponds to the sum of each region's RMSE ($w_{i,j}$=1). This misfit function mitigates any over- or under-

weighting of particular ocean regions that may arise from the irregular basinal distribution of observations (Fig. 1b). We refer to the immanent weights that arise for each ocean region from the application of BasinRMSE as basin-weighting, although the weighting is implicit and results from summing RMSEs with different numbers of observations. Our definition of regional constraints differs from that applied by Weber et al. (2018), since they defined nine discrete regions and incorporated only a portion of the observations provided in the GEOTRACES IDP2017.

The misfit functions based on the absolute residuals are referred to as RMAE (root mean absolute error) and BasinRMAE. The weights or number of regions applied are equivalent to those in the corresponding misfit functions based on squared residuals described above.

### 2.3.3 Synthetic observational constraints

For experiments listed in the first row of Fig. 1a (TWIN_ALL and synObs_ALL), the model is optimised towards synthetic

observational fields at the resolution of the model, i.e. residuals are calculated at all model grid points. In all other experiments, in the second and third rows of Fig. 1a, (synObs_IDP+, TWIN_IDP & and synObs_IDP), model output and observations are compared at the 3-D coordinates of the Zn observations in the IDP2017 (IDP experiments) or the extended version thereof (IDP+ experiments; Fig. 1b), which includes data from recent high-latitude studies not included in the IDP2017 (Sieber et al., 2019; Vance et al., 2019; Wang et al., 2019; Lemaitre et al., 2020). We only consider locations of IDP2017 data that were

assigned quality flags 1 or 2, indicating (probably) good quality and to which it is possible to interpolate. This results in ~4700 data points at 295 geographic locations to constrain the model in the IDP experiments. Relative to its fractional volume, the Atlantic is clearly over-represented in the IDP2017 relative to the other ocean basins (Fig. 1b), while the Indian Ocean and the Southern Ocean south of 50° S are under-represented. In the vertical, intermediate water depths are under-represented relative to the model's grid-spacing.

**3 Results and discussion**

Our ensemble of optimisations towards synthetic observations allows us to assess the influence of (i) uncertainty in data or biases in the model, (ii) data coverage, and (iii) misfit function on the ability of CMA-ES to reproduce biogeochemical model behaviour and parameter values. In the following, we first discuss the degree to which model parameter values could be constrained over all in our optimisation experiments, before discussing the influence of each of the above-mentioned aspects

in turn.



### 3.1 Parameter value retrieval and its sensitivities

Our TWIN experiments are a test case in which the model can exactly reproduce the synthetic observations. In these experiments, all parameter values were perfectly retrieved regardless of data coverage, even though calculating misfits in TWIN_IDP (i.e. with the data coverage of IDP2017) only involves 12% of the data from the target field of TWIN_ALL (perfect

data coverage). Thus, the reduced and inhomogeneous spatial coverage of the GEOTRACES IDP2017 will not prevent the optimisation algorithm from converging to the correct parameter values if the observations can be perfectly matched by the model equations. Figure 3 shows the evolution of the parameter values and the logarithmic misfit during the TWIN_ALL experiment. High variances are associated with a wide range of parameter values in a single generation of 10 individual simulations, and occur mainly at the beginning of the optimisation. For parameter $a_{Zn}$, which determines the asymptotic Zn:P

value of the non-linear portion of the Zn uptake equation (Eq. 2; Fig. 2), the average parameter value approaches the reference value earlier than for the other parameters.

In synObs experiments, i.e. when target fields cannot be perfectly reproduced by the model, CMA-ES does not exactly reproduce reference parameter values. However, optimisation almost always identifies a parameter set that gives a better fit to the target field than would have been produced with reference parameter values (Table S2). Figure 4 provides an overview of

our optimisation ensemble results in terms of the range of values determined for each of the four biogeochemical parameters optimised. This overview shows that the various types of complexity we introduce into our synObs experiments lead to a range of optimised values for each parameter. While the optimised values for parameter $b_{Zn}$ and $c_{Zn}$ span (almost) the entire range of allowed values, those for parameter $a_{Zn}$ and – to a lesser extent – $L$ span a relatively limited range (Fig. 4). Also, the median values for $a_{Zn}$ and $L$ lie close to the reference values, whereas the median value for $b_{Zn}$ is clearly higher than the reference

value, and that for $c_{Zn}$ coincides with the lower boundary. The fact that the optimised value for $c_{Zn}$ was found at its lower boundary in 70% of synObs experiments leads to an interquartile range in Fig. 4 that appears relatively narrow, although its optimised values range over the entire allowed parameter space, indicating the difficulty of constraining this parameter.

### 3.1.1 Interrelationship between parameter retrieval and model sensitivity

The differing constrainability of parameters may be understood in light of their influence on simulated Zn uptake systematics.

Figure 2 illustrates the extent to which each parameter affects the systematics of the stoichiometric uptake ratio $r_{Zn:P}$ when parameters are changed by ±50 % of the reference values, or when they are set to the minimum or maximum boundary value. While 50%-changes in the parameter value of $a_{Zn}$ have a relatively high impact on $r_{Zn:P}$, changing any of the other parameters by ±50 % affects the shape of the curve to a much smaller extent. Changes in $c_{Zn}$ mainly affect $r_{Zn:P}$ at high $Zn^{2+}$ concentrations, while changes in the parameters $b_{Zn}$ and $L$ have a similar effect on the shape of the $r_{Zn:P}$ curve, suggesting that

they might be able to compensate for each other.

The sensitivity of the system response to parameter $a_{Zn}$ has already been reported by de Souza et al. (2018). Indeed, our optimisation ensemble reveals that parameter $a_{Zn}$ has the strongest influence on simulated Zn export flux, especially in the





Southern Ocean (Fig. S2), where the Zn:P ratio of export plays an important role in determining the large-scale Zn distribution (Vance et al., 2017; Roshan et al., 2018; Weber et al., 2018). The sensitivity to parameter $a_{Zn}$ is also manifested in our

optimisations by the fact that this parameter generally converges first towards its optimised value (e.g. Fig. 3). Figure 5 and Table S2 provide information on the optimised parameters for each synObs experiment, i.e. parameter values that result in the minimum misfit of the last iteration. As illustrated clearly in Fig. 5, the reference value of $a_{Zn}$ is generally well retrieved (to within ±~30%; Table S2), with the important exception of VolRMSE-optimised synObs_circ solutions, where $a_{Zn}$ was found at its lower boundary, producing uptake systematics and global export fluxes that are clearly distinct from all others (Sect.

285  3.2.2).

Parameter $c_{Zn}$ represents the opposite case in terms of sensitivity. Coming into play only at high $Zn^{2+}$, its role in determining Zn uptake is minimal at the global scale, especially when high values of $a_{Zn}$, as in our reference parameter set, allow elevated $r_{Zn:P}$ at high latitudes. Thus, high values of $a_{Zn}$ decrease the importance of the linear portion of the uptake curve governed by parameter $c_{Zn}$ (Eq. 2) and the degree to which it is constrainable. The interaction between these two model parameters is

exemplified by the VolRMSE-optimised experiments in which $a_{Zn}$ is found at its lower boundary: here, optimisation finds elevated $L$ and extremely high values of $c_{Zn}$ (Fig. 5), coinciding with the upper boundary for this parameter. This results from the fact that high values of $c_{Zn}$ are needed to produce elevated Zn uptake at high latitudes when $a_{Zn}$ is low, especially when high ligand concentrations $L$ depress $Zn^{2+}$. In our ensemble of optimisations, high values of $c_{Zn}$ are always concomitant with elevated ligand concentrations (Figs. 5, S3).

Interdependence of parameter sensitivity can also be observed between the parameters $b_{Zn}$ and $L$. Although changes of opposite sign to these parameters produce similar changes in Zn uptake systematics (Fig. 2), both were correctly retrieved in our TWIN experiments (Table S2). In synObs experiments, underestimation of one of these parameters did not necessarily result in overestimation of the other. Exceptions to this are found when $L$ is greatly overestimated (>100%); in all these cases, $b_{Zn}$ is always clearly underestimated (<-26 %; Fig. S3). Higher values of the ligand concentration $L$ buffer $Zn^{2+}$ to

concentrations below typical values of $b_{Zn}$ over a large range of total Zn concentrations (Fig. S4), increasing the sensitivity of the uptake systematics to parameter $b_{Zn}$. Conversely, low values of $L$ result in a sharper rise of $Zn^{2+}$ with total Zn (Fig. S4), reducing the scope for $b_{Zn}$ to influence the Michaelis-Menten term in Eq. 2. In the vicinity of the reference parameter value of $L$, the sensitivity to parameter $b_{Zn}$ is relatively low (Figs. 2, S4). In our optimisations, this lack of sensitivity is manifested by the fact that $b_{Zn}$ is found at a boundary more frequently (>30 % of synObs experiments) than $L$ (<10 %). In our optimisations,

if parameters were found at one of their boundary values, we frequently observed that misfits lower than that produced with optimised parameters would have been achieved with parameter values outside of the prescribed boundaries. This finding supports previous studies suggesting that convergence of parameters to their prescribed boundaries, and the occurrence of lower misfits outside the prescribed and supposedly realistic parameter space, may point to deficiencies in biogeochemical model structure, wrong choice of parameters to be optimised, or bias in the physical circulation (e.g. Kriest et al., 2017; Falls

et al., 2021), and highlights the importance of well-considered boundaries for interpretability of results.



In summary, our results show that parameters with a stronger influence on the (reference) biogeochemical model behaviour are better constrained over the range of uncertainties and data-coverage limitations represented by our synObs experiments. Given our choice of reference parameters, which emphasises the high affinity Zn uptake system (non-linear term of Eq. 2), parameter $a_{Zn}$ is the best-constrained parameter, and $c_{Zn}$ the most difficult one to constrain, with its optimised value frequently
found at a boundary.

## 3.2 Retrieval of biogeochemical model behaviour

Figure 6 illustrates how the optimised parameter sets influence a key behaviour of the Zn cycling model: the dependence of the stoichiometric uptake parameter $r_{Zn:P}$ on dissolved Zn (hereafter *uptake curve* or *uptake systematics*). The uptake curve can be considered as a measure of similarity between optimised results in terms of the biogeochemical behaviour of the model.
In subsequent subsections, we describe the retrieval of reference biogeochemical model behaviour for three optimisation experiments with varying degrees of dissimilarity between optimised and reference uptake systematics.

### 3.2.1 RMSE-optimised synObs_ALL_seas

Experiment synObs_ALL_seas optimises our annual-mean model towards a target field produced when seasonal variability is simulated, with perfect data coverage. The RMSE-optimised parameter values in this experiment differ by 12–60 % from the
reference values used to produce the target field (Table S2), These values result in Zn uptake systematics that are broadly similar to the reference uptake systematics, although $r_{Zn:P}$ underestimates the reference $r_{Zn:P}$ at low concentrations, and exceeds it for all concentrations relevant to the surface ocean above ~1 nM (Fig. 6b). As a consequence, the RMSE-optimised Zn export flux is increased in the Zn-rich Antarctic Zone, but decreased in the Subantarctic Zone and at lower latitudes, relative to the reference Zn export flux distribution (Fig. S5b). These systematic changes reflect the trends observed when comparing
Zn export flux distributions in the target and reference simulations (Fig. S5d), although differences between the reference and the RMSE-optimised Zn export fluxes are smaller than those between the reference and the target. Similarly, a comparison of residuals in the Zn field of the RMSE-optimised model (Fig. 7b, e, h) to those between the reference and target field (Fig. S1e, h, k) reveals that optimisation has reduced the magnitude of residuals by up to ~50 %, while the patterns of the residuals remain near-identical. In both cases, the surface ocean simulated with annual-mean TMs is generally biased to higher Zn
concentrations (Figs. 7b, S1e). High positive residuals in the surface Southern Ocean and North Pacific are associated with negative residuals below the euphotic zone (e.g. Fig. 7e, h).

Although there are several optimisations resulting in a similar uptake curve as the RMSE-optimised synObs_ALL_seas experiment (Fig. 6), we would like to note that both distribution and magnitude of residuals can be quite different (cf. second columns of Figs. 7, 8) between experiments.





### 3.2.2 VolRMSE-optimised synObs_ALL_circ


In synObs_circ experiments, the target field was produced using a different physical model than that used during optimisation. The VolRMSE-optimised parameter values in synObs_ALL_circ, as well as those obtained in the corresponding optimisation with reduced data coverage, coincide with boundary values for three of four parameters (Table S2, Fig. 5). The resulting convex uptake curve (Fig. 6c) is strikingly different from the reference curve, as a consequence of a low value of $a_{Zn}$ and high

value of $c_{Zn}$. The VolRMSE-optimised parameter set results in extremely low global Zn export fluxes (Table S2), reducing Zn uptake to the extent that surface concentrations are not drawn to low values. This produces a positive bias throughout the surface ocean, and especially high concentrations in the subantarctic Southern Ocean relative to both the RMSE-optimised Zn (Fig. 8c) and the reference field. As a consequence, the VolRMSE-optimised model produces high Zn concentrations in the deep North Atlantic (Fig. 8f) and lower concentrations in the entire mid-depth to abyssal Pacific (Fig. 8i), i.e. a reduced deep-

ocean Zn gradient. These low Zn export fluxes reduce the normalised standard deviation of the VolRMSE-optimised field to a value similar to that in the target field (Fig. S6), i.e. VolRMSE-optimisation uses biogeochemical parameters to produce a similar statistical distribution of Zn as that simulated by MITgcm-ECCO due to its differing deep ocean circulation. The clearly distinct optimisation results are mainly related to volume-weighting, which is further discussed in Sect. 3.5.2.

### 3.2.3 RMAE-optimised synObs_IDP_circ

In the RMAE-optimised synObs_circ experiments, Zn uptake at high Zn concentrations is strongly reduced in synObs_IDP_circ relative to both the corresponding simulation with full data coverage and the reference uptake curve (Fig. 6c, f), resulting in globally higher surface-ocean Zn concentrations. Additionally, reduced surface Southern Ocean nutrient uptake and export decreases concentrations in deep waters of the Southern Ocean through reduced nutrient trapping (Sarmiento et al., 2004; Marinov et al., 2006; Primeau et al., 2013), and, because reduced Zn uptake is a consequence of a decrease in the

uptake stoichiometry parameter $r_{Zn:P}$, it leads to a strengthening of the global Zn-PO$_4$ correlation while the Zn-Si correlation is weakened (Vance et al., 2017; de Souza et al., 2018). The shape of the RMAE-optimised uptake curve in synObs_IDP_circ is unique in our optimisation ensemble (Fig. 6), and is a consequence of the joint effect of reduced data coverage and choice of misfit function, as we discuss in Sect. 3.6.

### 3.3 Influence of uncertainty in the target field on parameter retrieval

In our synObs experiments, the model cannot exactly reproduce the target field, and optimisation finds parameter sets that differ from the reference parameters to varying degrees. With increasing dissimilarity between target field and reference field, the reconstruction of model behaviour becomes increasingly difficult, and the sensitivity of optimisation results to the applied misfit metric increases (Fig. 6; Sect. 3.5). In all synObs experiments except one (Sect. 3.5.1), CMA-ES found a parameter set that produces a lower misfit to the target field than would have been achieved using the reference parameter values. This can

be seen as "reciprocal bias compensation", a term coined by Löptien and Dietze (2019) to describe the phenomenon that part



of the bias induced by flaws in circulation models can be compensated for by changes to biogeochemical parameters. We find such error-compensating effects induced by biogeochemical parameter optimisation in all our synObs experiments. Relative to the misfit obtained with the reference parameters, proportionally highest reductions in misfits are seen in synObs_seas experiments (up to 3.5 %, excluding VolRMSE; Table. S2).

### 3.3.1 Analytical uncertainty

In synObs_noise experiments, which aim to assess how CMA-ES is affected by analytical uncertainty inherent in any true observational field, model Zn uptake behaviour is relatively well reconstructed regardless of data coverage (Fig. 6a, d) except for VarRMSE-optimised synObs_IDP_noise, discussed further in Sect. 3.5.1. However, in contrast to the TWIN experiments, in which parameter values were perfectly retrieved, the optimised parameters in synObs_noise experiments are distinct from the reference parameters (Fig. 5): while parameters $a_{Zn}$, $b_{Zn}$, and $L$ are relatively well constrained to within ~5 %, ~30 % and ~15 % respectively (excluding VarRMSE), parameter $c_{Zn}$ is consistently found at its lower boundary. The poorer constraints on $b_{Zn}$ and $L$ than on $a_{Zn}$ may be explained by reciprocal effects of these parameters on the uptake curve (Fig. 2), while model sensitivity to $c_{Zn}$ is generally low when $a_{Zn}$ is properly reproduced (Sect. 3.1.1). More broadly, despite the difference in the type of data being used for optimisation, the imperfect parameter retrieval in these experiments is consistent with the observation in data-assimilating ecosystem model studies that Michaelis-Menten constants are hard to constrain in optimisations against synthetic data disturbed with noise (Friedrichs et al., 2006; Löptien and Dietze, 2015).

### 3.3.2 Lack of seasonal variability

In synObs_seas experiments, which aim to assess the impact of our model's lack of seasonality, differences between optimised uptake curves are more pronounced for different misfit functions than for different data coverage (Fig. 6b, e), with the VolRMSE-optimised uptake curve most obviously different from the reference curve. Nonetheless, in experiments with reduced data coverage, both the RMSE- and the VolRMSE-optimised uptake curves are less similar to the reference uptake curve than in corresponding experiments with full data coverage. In all synObs_ALL_seas experiments, the surface ocean of the optimised model is generally biased to higher Zn concentrations (e.g. Fig. 7b, c), accompanied by negative residuals below the euphotic zone (e.g. Fig. 7e, h; Sect. 3.2). The integrated Zn export fluxes obtained in our synObs_seas experiments are generally slightly lower than the reference flux, but always overestimate the export flux of the target simulation (Table S2; Fig. S9). The VolRMSE-optimised synObs_ALL_seas experiment produces the lowest export flux, i.e. closest to the flux of the target simulation, but this optimisation leads to residuals that tend to *amplify* the RMSE-optimised residuals (cf. second and third column in Fig. 7), demonstrating the importance of the spatial patterns in the Zn export flux. Figure S7 shows that it is the differences in both circulation and biogeochemistry that limit the ability of CMA-ES to reconstruct exact parameter values and the integrated Zn export flux underlying the target field (Table S2). Instead, optimisation finds a compromise solution that alters biogeochemical parameter values to compensate for systematic differences between the target and reference fields, i.e. reciprocal bias compensation sensu Löptien and Dietze (2019).



### 3.3.3 Differences in underlying circulation

In synObs_circ experiments, our model is optimised towards a target field which was created using the reference
biogeochemical parameters in a different circulation model, MITgcm-ECCO (Sect. 2.3.1). Among the target fields used in this
study, the target field produced with MITgcm-ECCO is most clearly distinct from the reference field (Figs. 1d, e, S1).
Differences between the reference field and the synObs_circ target field are larger than any differences resulting from relatively
large changes to Zn uptake systematics within the MITgcm-2.8 framework used during optimisation (Fig. S6). Despite clearly
different uptake curves (Fig. 6c), it remains the case that differences between the optimised Zn fields and the target are larger
than differences between optimised models resulting from different misfit functions (cf. second column and third column in
Fig. 8). Furthermore, differences between RMSE-optimised and VolRMSE-optimised fields, which are of a purely
biogeochemical origin within the same circulation framework (MITgcm-2.8), are much more systematic than the distribution
of residuals to the target field, which compares results from two different circulation frameworks (MITgcm-2.8 for the
optimised model, MITgcm-ECCO for the target field). A major difference between the circulation simulated by the two
MITgcm-configurations relates to timescales of deep ocean circulation, especially in the voluminous deep Pacific (Fig. S8).

The focus of our synObs_circ experiments is mainly to assess the effect of data coverage on optimisation results when a
systematic circulation bias exists, which we discuss in Sect. 3.4. However, it is worth noting that (i) parameter retrieval appears
most challenging in these experiments, with parameters $b_{Zn}$ and $c_{Zn}$ each converging to a boundary in 8 of the 11 experiments
conducted (Fig. 5), (ii) the synObs_circ experiments are the only experiments in which parameter $a_{Zn}$, the best-constrained
parameter (Sect. 3.1), converged to its lower boundary as a consequence of volume-weighting (Sects. 3.2.2, 3.5), and (iii) all
synObs_circ optimisations result in global Zn export lower than that of the reference simulation, while export in the target
simulation is 7% *higher* than in the reference simulation (Table S2, Fig. S9). The fact that retrieval of parameter values and
biogeochemical model behaviour is difficult in the face of systematic differences in whole-ocean ventilation timescales is not
unexpected. The strong influence of the physical circulation framework on biogeochemical ocean model output was
emphasised relatively early by Doney (1999). More recently, by comparing optimisation results obtained using CMA-ES with
TMs derived from three different ocean models, Kriest et al. (2020) found that some of their optimised biogeochemical
parameters depended strongly on the circulation.

In summary, we find that optimisation of biogeochemical parameter values introduces some error-compensating effects in all
synObs experiments. This tendency is most simply illustrated by the synObs_noise experiments, where optimisation to noise-
perturbed data produces parameter sets slightly different from the reference set. However, relative to the misfit obtained with
the reference parameters, highest misfit reductions are seen in synObs_seas experiments (up to 3.5 %, excluding VolRMSE;
Table. S2). This may reflect that fact that the target field of these experiments differs most strongly from the reference field in
the high latitudes (Fig. 7), where changes to parameter values have a relatively high impact.



### 3.4 Influence of reduced data coverage on parameter retrieval

In order to isolate the effect of reduced data coverage on optimisation, we compare the results of synObs_IDP(+) experiments, in which data coverage is limited to the locations of existing observations, with the corresponding synObs_ALL experiment in which the entire Zn field is used to quantify misfit.

### 3.4.1 Effects of reducing data coverage

Limiting simulated data coverage to that of the IDP2017 means that approximately 88% of the model cells are not used for
comparison to synthetic observations. Nonetheless, this does not lead to a significant increase in the total number of iterations needed until the internal convergence criterion is reached (Table S2). We also found no evidence that the reduction in data coverage causes CMA-ES to terminate in a local minimum: calculating the misfit with the optimised model output from synObs_ALL experiments at IDP2017 coordinates results in a higher misfit than the minimum misfit achieved in the corresponding synObs_IDP experiment. Conversely, misfits calculated using all model cells with optimal parameters from the
synObs_IDP experiments were higher than the corresponding minimum misfits in synObs_ALL. Thus, the objectively optimal parameters indeed depend on the data coverage of the target field.

Figure 6 shows that Zn uptake behaviour obtained with a particular misfit function in synObs_IDP often did not greatly differ from that obtained in the corresponding synObs_ALL experiment – with the exception of the RMSE-optimisation with the seasonal target field (synObs_seas) and the RMAE-optimisation with the target field obtained from ECCO (synObs_circ),
introduced in Sect. 3.2. However, Fig. 9 shows that the degree of differences in parameter values varies: while for parameter $b_{Zn}$ optimal solutions scatter widely around the 1:1 line, differences for the other parameters are much less pronounced, with only a few major offsets. The similarity for parameter $c_{Zn}$ is a consequence of its frequent coincidence with a boundary (Fig. 5), regardless of data coverage. In contrast, the smaller scatter of parameters $L$ and especially $a_{Zn}$ around the 1:1 line reiterates our finding in Sect. 3.1 that these parameters are better constrained by optimisation (Fig. 4). While the scatter for parameter $L$
is about equally distributed around the 1:1 line, parameter $a_{Zn}$, which has the strongest effect on Zn uptake systematics and export flux (Figs. 2, S2), is always underestimated in synObs_IDP experiments relative to the value obtained in synObs_ALL. Limiting data coverage reduces the optimised parameter value of $a_{Zn}$ by $7 \pm 5$ % (1SD), excluding two cases for which $a_{Zn}$ is underestimated by >30 %: (i) VolRMSE-optimised synObs_IDP_seas, and (ii) RMAE-optimised synObs_IDP_circ (Fig. 9a, Table S2). In the first case, underestimation of $a_{Zn}$ is partly compensated by underestimation of parameters $b_{Zn}$ and $L$,
producing an uptake curve not very different from that obtained in in the corresponding synObs_ALL (Fig. 6b, e), and similar integrated Zn export (Fig. S9). In contrast, for RMAE-optimised synObs_IDP_circ, the underestimation of parameter $a_{Zn}$ is not compensated by other parameter values, leading to very different Zn uptake systematics from the corresponding synObs_ALL experiment (Fig. 6c, f) and strongly reduced integrated Zn export (Fig. S9). A similar but less extreme example leading to different Zn uptake systematics from the corresponding synObs_ALL experiment is found in the RMSE-optimised
synObs_IDP_seas experiment, in which underestimation of $a_{Zn}$ is compounded by overestimation of $L$ and $c_{Zn}$ (Fig. 6b, e).





It is apparent from Figs. 6 and 9 that the extent to which data coverage reduction affects parameter retrieval depends on the misfit function; we discuss this in detail in Sect. 3.6. Given the appropriate choice of misfit function, however, our set of experiments indicates that the spatial coverage of the GEOTRACES IDP2017 is sufficiently representative of the large-scale patterns of the Zn distribution to allow retrieval of biogeochemical model behaviour through optimisation. With regard to

parameter retrieval, this finding is limited to those parameters that dominate model behaviour, i.e., $L$ and, especially, $a_{Zn}$, although the latter is consistently slightly underestimated relative to the value obtained in the corresponding synObs_ALL experiment. The fact that the generally well-constrained parameter $a_{Zn}$ is underestimated when data coverage is reduced, even though this parameter sensitively controls Zn export from the biogeochemically-important Southern Ocean (Fig. S2b), motivates an assessment of whether increasing the observational density in some regions, particularly the Southern Ocean,

may improve parameter retrieval. We thus subsequently assess potential benefits associated with the addition of high-latitude data by considering our synObs_IDP+ experiments in more detail.

### 3.4.2 Effect of including high-latitude data

The synthetic target fields of our synObs_IDP+ experiments complement the IDP2017 coordinates with the locations of high-latitude Zn observations published more recently (Fig. 1b). For the two optimisation experiments discussed above (RMSE-

optimised synObs_seas and RMAE-optimised synObs_circ), including these additional constraints leads to an improvement in parameter retrieval, as the underestimation in $a_{Zn}$ is considerably reduced (by ~50 %; Fig. 9), and the optimised uptake systematics become more similar to those obtained in the corresponding synObs_ALL experiments (Fig. 6). In order to investigate whether the alignment in systematics of the synObs_IDP+ experiments with those of the corresponding synObs_ALL experiments comes about due to additional Zn-concentration information, or due to the implicit change in the

weights of different oceanic regions resulting from these additional data, we carried out an additional RMAE-optimised synObs_IDP_circ optimisation in which each data point was weighted by the implicit basin-weights derived from the IDP+ data situation (Table S3). This optimisation, which only includes the Zn information resulting from IDP2017 data coverage, resulted in uptake systematics that are quite distinct from those obtained in the corresponding synObs_IDP+ experiment. The uptake curve is in fact very similar to that obtained in synObs_IDP_circ (Fig. S10), but produces consistently higher values of

$r_{Zn:P}$. This similarity implies that rather than the changed weighting of oceanic regions, it is the additional Zn-concentration information in the IDP+ data that is mainly responsible for altering optimisation results.

Further evidence for the importance of the additional high-latitude Zn data arises from the BasinRMAE-optimised uptake systematics in synObs_IDP+_circ, which becomes more similar to the corresponding synObs_ALL experiment than that obtained for synObs_IDP_circ (Fig. S10). We suppose that in particular the additional data constraints at southern high

latitudes, which are typically associated with high Zn surface concentrations, can significantly improve the ability to constrain biogeochemical model behaviour, because (i) the largest differences between the RMAE-optimised uptake systematics obtained in synObs_IDP_circ and synObs_IDP+_circ are observed for high Zn, and (ii) the BasinRMAE-optimised uptake



curve obtained in synObs_IDP+_circ improves the BasinRMAE-optimised uptake curve obtained in synObs_IDP_circ, again exceeding it especially at high Zn (Fig. S10).

In summary, our results suggest a benefit from additional high-latitude data. Among the RMAE-based optimisation results with reduced data coverage, BasinRMAE-optimised uptake systematics in synObs_IDP+_circ best reproduce the RMAE-optimised uptake systematics in the corresponding experiment with full data coverage. This finding, together with the smaller deviations from the 1:1 line achieved with basin-weighting misfit metrics in Fig. 9, suggests that basin-weighting may be advantageous in reconstructing biogeochemical system behaviour from sparse and inhomogeneously sampled data. We discuss

this in Sect. 3.6.

## 3.5 Importance of misfit function

A feature that emerges from the preceding discussion is that, both in the presence of inaccuracies in the physical model (Sect. 3.3) or with a reduction in data coverage (Sect. 3.4), optimisation results become increasingly dependent on the misfit metric used. This is in agreement with previous studies that report a potentially large impact of the choice of misfit function on the

best estimate of biogeochemical fluxes and concentrations (e.g. Evans, 2003; Sauerland et al., 2019). In particular, the subjective choice of weights may have a strong influence on the optimisation results (Evans, 2003). While some studies apply RMSE-based misfit functions (e.g. Friedrichs et al., 2007), others suggest to reduce the effect of outliers by using a misfit function based on the absolute differences between model and observations (Trudinger et al., 2007; Seegers et al., 2018). Our ensemble of optimisations suggests that weighting is more important than the choice of squaring the residuals or not. Our

optimisation results show that weighting squared residuals by the fractional volume of the grid cell (VolRMSE) results in large deviations from the reference uptake systematics whenever systematic differences in the underlying physical model are present (i.e. in synObs_seas and synObs_circ experiments), even with perfect data coverage, and weighting squared residuals by the inverse variance prevented CMA-ES from reaching its internal termination criterion. Other examples of a misfit metric leading to poorly retrieved biogeochemical model systematics emerge only in the light of both underlying uncertainty *and* limited data

availability, and will be further discussed in Sect. 3.6, where we elucidate the joint effect of these two aspects and the importance of basin-weighting.

### 3.5.1 Influence of misfit function: variance-weighting

The only synObs_noise experiment entirely unable to reproduce model Zn uptake behaviour is that optimised using VarRMSE (Fig. 6d). Reasons for CMA-ES being prevented from finding the absolute minimum of the misfit function might be non-

informative ("flat") misfit functions, misfit functions with a rough topography, too wide boundary constraints, or a small population size (e.g. Ward et al., 2010; Kriest et al., 2017).The VarRMSE misfit metric is highly sensitive to changes in parameter $b_{Zn}$ (Fig. S11), and its minimum in parameter space is strongly offset from the reference value of this parameter. This topography appears to be a consequence of the variance-weighting of residuals in this misfit function. Some of the weighted residuals between the reference field and the noise-perturbed target field are very high for small concentrations (Fig.





S12). This is because the variance used for weighting the squared residuals is empirically estimated from the noise-perturbed observations, rather than from the reference field used to determine the added noise (Sect. 2.3.1). Since $\varepsilon$ increases rapidly at small concentrations (Eq. 3), the empirical variance determined for noise-perturbed observations that actually underestimate the true value is a gross underestimate, leading to *over*weighting of these residuals in the misfit function (since $w_{i,j} = \varepsilon^{-1}$). As a consequence, this particular misfit function appears to be too sensitive to changes in parameter $b_{Zn}$, which plays an important

role at low concentrations (Fig. 2). However, the sensitivity of the *global* Zn distribution to parameter $b_{Zn}$ is virtually negligible (de Souza et al., 2018), and thus an elevated sensitivity of the optimisation to this parameter is not desirable. It might be sensible to apply a minimum absolute error when calculating weights if using a misfit metric that weights by its reciprocal (Schartau et al., 2001).

### 3.5.2 Influence of misfit function: volume-weighting

A striking result of our optimisation ensemble is the fact that, in the presence of biases in the model, VolRMSE-optimised solutions exhibit Zn uptake systematics that are most distinct from both the other experiments as well as the reference uptake curve (Fig. 6), even with perfect data coverage. The VolRMSE misfit function compensates to some extent for the unequal distribution of model cells, which are more numerous in the upper ocean due to the higher vertical resolution there. On the other hand, all misfit metrics applied in this study naturally emphasise the deep ocean for a nutrient element such as Zn, due

to the order-of-magnitude increase in Zn concentrations between the surface and the deep ocean. Because of this expected concentration dependence of the residuals, several studies suggest that log-transformation might be appropriate if there is such a wide variability in concentrations (e.g. Stow et al., 2009; Seegers et al., 2018; Falls et al., 2021); instead, VolRMSE exacerbates this concentration dependence for elements with a nutrient-like distribution. When there are differences in the underlying physical model (synObs_seas and synObs_circ), VolRMSE-optimisation consistently leads to the smallest globally

integrated Zn export flux (Table S2). This finding is most strongly manifested in synObs_circ experiments, in which there are systematic differences between the large-scale circulation underlying the synthetic observations (from MITgcm-ECCO) and the model (MITgcm-2.8).

The biogeochemical model behaviour obtained with the VolRMSE misfit metric in synObs_ALL circ has been introduced in Sect. 3.2. In this experiment, the optimised parameter set results in integrated Zn export flux reduced by 15 % relative to the

reference simulation (Table S2), and by 20 % relative to the MITgcm-ECCO simulation that produced the target field; the effect is even stronger for the corresponding experiment with IDP2017 data coverage. The fact that VolRMSE-optimised uptake curves strongly deviate from the reference uptake curve even with perfect data coverage (Fig. 6b,c), while only limited deviations are seen with RMSE and RMAE, supports the suspicion of Kriest (2017) and Kriest et al. (2017) that volume-weighting might impede determinacy of parameters related to processes taking place in the euphotic zone. Based on our results,

we question the suitability of volume-weighting for optimisation of parameters related to biological uptake towards basin-scale dissolved data, because VolRMSE leads to the fitting of large-scale patterns associated with ventilation of the deep ocean, rather than fitting biogeochemical model behaviour associated with the parameters to be optimised. This is shown particularly





clearly by the synObs_circ experiments discussed above: simulating ideal age (Thiele and Sarmiento, 1990), the physical model used during optimisation (MITgcm-2.8) produces deep waters that, especially in the Pacific, are significantly older than

in the circulation model underlying the synthetic target field (Fig. S8). This large-scale circulation timescale difference leads to enhanced accumulation of regenerated Zn in the deep Pacific relative to the target field (Fig. 8). The use of VolRMSE results in a sensitivity to residuals in the Zn-rich deep ocean to such an extent that the misfit minimum is found for parameters that drastically decrease globally integrated Zn export in order to reduce Zn accumulation in old deep waters – even though this simultaneously results in unrealistically high surface-ocean Zn concentrations (Sect. 3.2.2). On the one hand, these results

recapitulate the dependence of biogeochemical model results on physical circulation pathways and timescales, but they also reveal that such sensitivities may be exacerbated by the sensitivities of the misfit function chosen for optimisation. Our results suggest that VolRMSE tends to enhance the circulation-dependence of optimisation results, although this tendency may be strengthened in our study by the nutrient-restoring nature of the underlying P-cycling model (e.g. Kriest et al., 2020).

### 3.6 Interaction between data distribution and misfit function: importance of basin-weighting

Finally, we bring together the two aspects of our raster of optimisation experiments discussed in Sects. 3.4 and 3.5, in order to assess how the influence of a misfit function on optimisation results is affected by data distribution. In particular, we discuss the importance of basin-weighting when data coverage is reduced.

The two experiments in which the reduction in data coverage induces relatively large differences from those obtained in corresponding synObs_ALL experiments are the RMSE-optimised synObs_IDP_seas experiment and the RMAE-optimised

synObs_circ experiment (Table S2; Fig. 6). Figure 10 shows the depth distribution of the residuals in these two experiments, as well as the residuals between the reference field and the corresponding target field. The depth distribution of residuals between the reference field and the target field of the synObs_seas experiments reveals that the sum of squared residuals in the surface ocean is about as high as that in the abyssal ocean, when data coverage is perfect (Fig. 10b). However, restricting data coverage to IDP2017 coordinates leads to a larger normalised sum of residuals (and especially of squared residuals) in

the abyssal ocean, simply due to the sampling locations at which residuals are calculated. These residuals are reduced by optimisation in the synObs_ALL and synObs_IDP experiments, but the depth structure of the residuals, and the enhanced importance of the abyssal ocean with IDP2017 coverage, remains (Fig. 10b). Overall, misfit minimisation appears to be a trade-off between fitting Zn concentrations in the abyssal ocean and that in the uppermost ~300 m: with full data coverage, RMSE-optimisation slightly favours fitting the upper ocean, but with reduced data coverage, changes to Zn concentrations in

the abyssal ocean affect misfit more strongly than the surface ocean.

The analysis of the depth distribution of residuals between the reference field and the target field of the synObs_circ experiments (Fig. 10c, d, h, i) reveals that limiting data coverage to IDP2017 coordinates increases normalised residuals between reference and target field throughout the water column (Fig. 10c, d). The RMAE-optimised synObs_ALL_circ experiment produces a Zn field that is almost identical to the reference field, while in the corresponding experiment with

IDP2017 coverage, optimisation worsens model fit in the uppermost layers (Fig. 10i) in favour of reduced sums of absolute



residuals at depth, a result that is promoted by the small magnitude of residuals in the Zn-poor upper ocean. Figure S9 suggests that nearly all synObs_IDP_circ optimisations achieve lower global misfit at the expense of increased surface residuals produced by a decreased global Zn export flux. Thus, reducing data coverage for optimisation can alter misfit trade-offs between different ocean regions, due to changes in the vertical or geographic sampling of the target distribution.

In order to assess the joint effect of reduced data coverage and misfit function on optimisation results in more detail, Fig. 11 compares the optimised Zn obtained in synObs_IDP_circ experiments with that obtained in synObs_ALL_circ. The synObs_IDP_circ experiments are the experiments with the greatest variability among optimised uptake systematics (Fig. 6), and the most-influential and best-constrained parameter $a_{Zn}$ is consistently underestimated, relative to the optimised value obtained in the corresponding synObs_ALL_circ experiment (Fig. 9; Sect. 3.4), a finding that is especially clear with the

RMAE misfit metric.

The comparison of the RMAE-optimised Zn fields obtained from synObs_ALL_circ and synObs_IDP_circ against each other, either only at IDP-coordinates (Fig. 11a) or at all model grid points (Fig. 11b), reveals that IDP2017 sampling does indeed capture many of the systematic offsets between the two fields, such as those at high concentrations in the Pacific and Southern Ocean >50° S. However, the offsets at IDP2017 coordinates are apparently neither numerous nor large enough to drive the

RMAE-optimisation towards the result obtained with full data coverage. Squaring residuals amplifies the relative impact of these offsets (Fig. 11c, d), and indeed a comparison of the corresponding RMSE-optimised Zn field to RMSE-optimised synObs_ALL_circ reveals that such offsets between the two simulations are virtually absent at IDP coordinates (Fig. 11e). The fact that the RMSE-optimised $r_{Zn:P}$ obtained in synObs_IDP_circ underestimates that in synObs_ALL_circ at high concentrations but slightly overestimates it at low concentrations (Fig. 6c, f) results in tiny but systematic deviations from the

theoretical quadratic relationship in the Southern Ocean (>50° S) and the Indian Ocean (grey points in Fig. 11e), which appear to be barely captured with IDP coordinates.

**Basin-weighting.** Another way to amplify systematic offsets seen in Fig. 11a is achieved by calculating separate RMAEs for different ocean basins as in our basin-weighting scheme, which weights the differences between the two fields in the Southern Ocean >50° S more strongly (Fig. 11f). The systematic Zn differences in this region resulting from different data coverage are

mostly mitigated in the corresponding BasinRMAE-optimised simulation, which produces a field much more similar to RMAE-optimised synObs_ALL (Fig. 11h) and results in estimates of $a_{Zn}$ and $L$ that are less strongly offset from this solution (Fig. 9; Table S2). Similar alignments in parameters $a_{Zn}$ and $L$ with basin-weighting can be observed in the case of the RMSE-optimised synObs_IDP_seas experiment discussed above. Here, the BasinRMSE-optimised solution corrects both the underestimation of $a_{Zn}$ and the overestimation of $L$ (Fig. 9a, d; Table S2) and produces uptake systematics more similar to the

corresponding synObs_ALL solution (Fig. 6b, e). Thus, in both these experiments, basin-weighting reduces the sensitivity of optimisation results to the data distribution. Our basin-weighting scheme was chosen to counteract the unequal distribution of observations between basins in the IDP2017 (Fig. 1b). Thus, basin-weighted metrics result in a relative down-weighting of the Atlantic and the Pacific, while weights of the Indian Ocean and the Southern Ocean south of 50° S are increased. As discussed for the two examples above, improvements achieved through basin-weighting are more generally apparent in our ensemble,





being reflected in approximation of the uptake systematics in synObs_IDP(+) experiments to those in the corresponding synObs_ALL experiments with perfect data coverage (Fig. 6), and in a closer correspondence between the numerical values of optimised parameters $a_{Zn}$ and $L$ (Fig. 9).

It is worth noting that our experience with basin-weighting is to some extent in contrast with the findings of Tjiputra et al. (2007) who applied variational data assimilation to a three-dimensional global marine biogeochemical model. Calculating

misfits from surface cells only, these authors found that twin experiments aiming to assimilate synthetic chlorophyll "observations" are more successful in reducing the misfit function if an implicit regional scaling is applied by weighting each residual by its fractional volume, i.e., a metric similar to VolRMSE, which performs poorly in our experiments (Sect. 3.5.2). On the other hand, some optimisations towards the distributions of dissolved nutrients (e.g. Frants et al., 2016) explicitly encode an ad-hoc emphasis of the Southern Ocean that is implicit in our basin-weighting scheme. Though basin-weighting

might be considered subjective, we argue that the high zonal symmetry in the Southern Ocean and its key role in determining global ocean nutrient distributions (Sarmiento et al., 2004; 2007) are sufficient justification for its application. In particular, we hypothesise that the biogeochemical importance of the Southern Ocean in determining the global Zn distribution (Vance et al., 2017; de Souza et al., 2018; Weber et al., 2018) is the reason why it is preferable to (implicitly) emphasise this region in the misfit metric. Given the general importance of the Southern Ocean in determining large-scale ocean biogeochemical

parameters, such a metric is likely to perform well for most biogeochemically cycled elements with long oceanic residence times. Nonetheless, different misfit metrics obviously capture different aspects of the distribution of model performance. In order to ensure a thorough skill evaluation, Stow et al. (2009) suggest that the use of several metrics simultaneously is often to be recommended, and Sauerland et al. (2019) show that multi-objective optimisation can help to better constrain model parameters.

**3.7 Implications for model calibration using real data**

In this study, we have separately assessed how optimisation results are impacted by different sources of uncertainty. In accordance with other studies (e.g. Löptien and Dietze, 2019; Kriest et al., 2020), our ensemble of optimisations shows that biogeochemical parameters are often optimised to compensate for the inability of model formulations to reproduce the target field (Sect. 3.3). The quantitatively largest error-compensating effects were seen in synObs_seas experiments. However, with

the exception of the VolRMSE-optimised experiments (Sect. 3.5.2) and a few experiments optimising against reduced data coverage (Sects. 3.4 and 3.6), the optimised parameters do not result in large deviations from the reference uptake curve (Fig. 6), such that the systematics of biogeochemical model behaviour are retrieved even if exact parameter values are not. Reconstructed Zn uptake systematics were most different from the reference uptake systematics in experiments with systematic differences between the large-scale circulation of the model and that underlying the target field (synObs_circ; Fig. 6); misfits

obtained in this experiment type were also about an order of magnitude higher than those for synObs_seas, which differ only in terms of the presence or absence of seasonality within the same physical model. Because increasing both spatial and temporal resolution might be computationally unaffordable, even for a relatively efficient global optimisation algorithm like CMA-ES,



we suggest that for studies focusing on global (micro)nutrient distributions with long whole-ocean residence times, it is important to prioritise the choice of circulation model, with special focus on accurate simulation of large-scale circulation
timescales.

More generally, our optimisation ensemble suggests that a misfit function that appears suitable for optimisation in a simple TWIN experiment, in which the model can perfectly describe the target field, may not be the best choice for optimisation towards noisy, incomplete and/or irregularly distributed real-world data. Specifically, it is important to recognise the subjectivity that the choice of misfit function introduces to objective parameter optimisation, and to carefully weigh the
sensitivities implicit to the misfit function in making this choice for any particular application. Although computationally more demanding, multi-objective optimisation (e.g. the multi-objective CMA-ES of Sauerland et al., 2019) might present a useful approach to comparing the differing parameter sensitivities of, and interrelations between, misfit functions in a particular optimisation problem.

We also emphasise that any calibration towards real data will also be influenced by factors that were not relevant in this study.
Firstly, all synthetic observations used herein were created using a biogeochemical model with the same functional form for Zn uptake (Eqs. 1, 2) as that encoded into the model. While observations from wild phytoplankton (Twining and Baines, 2013) indicate geographical systematics that are similar to those that result from this model formulation (de Souza et al., 2018), there is no reason to believe that the stoichiometry of Zn:P uptake in the real ocean should follow a single dependence on dissolved Zn concentration; thus, any optimisation of a simple biogeochemical model such as ours towards real data must be seen as
attempting to retrieve the systematics of biogeochemical behaviour, rather than physically meaningful parameter values. This reasoning underlies our choice to discuss the results of our optimisation in the context of Zn uptake systematics (Fig. 6) and large-scale export flux. Secondly, our study has not considered the influence of simplifications, e.g. in the underlying P cycling model. In our model formulation, P cycling directly affects Zn cycling, as Zn uptake is related to $PO_4$ uptake through $r_{Zn:P}$ (Sunda and Huntsman, 1992), and since Zn is assumed to remineralise with the same globally constant length-scale as P
(Twining et al., 2014). Particularly the latter assumption may be over-simplified, as the remineralisation length-scale might be dependent on latitude or upper-ocean temperature (DeVries et al., 2014; Marsay et al., 2015; Weber et al., 2016). Furthermore, observational studies have come to contrasting conclusions regarding the similarities between the regeneration length-scales of Zn and P (Twining et al., 2014; Ellwood et al., 2020; Cloete et al., 2021).

## 4 Conclusions

This study has assessed how data distribution, model imperfections and misfit function influence the optimisation of a marine Zn cycling model with the algorithm CMA-ES. Using synthetic observations that allow us full control over the target field, we aimed to investigate the algorithm's skill at retrieving parameter values and emergent model behaviour under real-world conditions resulting from data constraints, such as reduced data coverage and analytical errors, or from systematic bias between model and target field related to either seasonality or large-scale physical circulation.



Our results revealed good performance of CMA-ES with respect to recovering biogeochemical model behaviour. In TWIN experiments, in which the model was optimised towards target fields that could theoretically be perfectly reproduced by the model, CMA-ES recovered all model parameter values regardless of data coverage. Furthermore, the analysis of our suite of synObs experiments, in which reproduction of reference model behaviour was impeded since the target field could a priori not be exactly reproduced by the model, revealed that (i) the data coverage of the GEOTRACES IDP2017 can be sufficient to

reconstruct the systematics of Zn cycling at the global scale, (ii) optimisation generally broadly reproduced the Zn uptake systematics of the reference simulation, with a few meaningful exceptions related to the choice of misfit function, and (iii) the degree to which a parameter can be constrained depends strongly on its influence on the model's Zn uptake systematics and emergent properties such as global export flux.

As CMA-ES generally identified parameter sets that produced lower misfits than would have been calculated with the reference

parameter set, all optimised results contain some error-compensating effects. Despite these, the reference Zn export flux is generally relatively well reproduced, except with the VolRMSE misfit metric. Applying this metric, which deemphasises the shallow ocean and polar regions, results in the most distinct Zn uptake systematics from both the reference curve and those resulting from optimisation with other misfit metrics. Furthermore, the inability of this metric to reproduce model behaviour increases with the dissimilarity between the target field and reference field (Fig. 6). Based on our results, we suggest avoiding

misfit metrics that deemphasise regions where parameters to be optimised are likely to be influential.

Finally, our study emphasises the importance of implicit basin-weighting in the misfit function, and the significance of the information gained from an increase in high-latitude Zn concentration data. The basin-weighting misfit metrics applied in this study (BasinRMSE and BasinRMAE) oppose differences in data coverage between basins, and prove most successful in minimising the sensitivity of optimised model behaviour to data coverage. Since the high latitudes are under-sampled in the

extant data, the efficacy of basin-weighting, in turn, reveals the importance of high-latitude Zn data for constraining model behaviour, as does the fact that our best-constrained parameter – which dominantly determines the magnitude of Zn export at high latitudes – is underestimated when data coverage is reduced.

**Code and data availability**

The TMM software and transport matrices are available to download from https://doi.org/10.5281/zenodo.1246300

(Khatiwala, 2018). The OptClimSO optimisation framework is available at https://doi.org/10.5281/zenodo.5517610 (Oliver, 2021). It is originally sourced from Tett et al. (2013), and includes the CMA-ES optimisation code taken from Kriest et al. (2017). Code implementing the Zn model optimised model outputs are available from https://doi.org/10.3929/ethz-b-000543389 (Eisenring, 2022).



### Author contributions

GFDS and CE conceived the study, and designed the experiments. SEO and CE developed code, SEO and SK advised on implementation. CE carried out all experiments and analysis, and wrote the first draft. GFDS and other authors contributed to the ideas presented in this study and provided input into the final manuscript.

### Competing interests

The authors declare that they have no conflict of interest.

### Acknowledgments


We thank Derek Vance for his helpful comments on an earlier version of this manuscript. This work was supported by a grant from the Swiss National Supercomputing Centre (CSCS) under project ID s941. CE is supported by Swiss National Science Foundation grant 200021_192116 to GFdS. SEO is supported by the National Environmental Research Council (NE/L002612/1), the Oxford Doctoral Training Partnership in Environmental Research and the Met Office.

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





**Table 1.** Reference parameter values and search intervals.

|  | $a_{Zn}$ | $b_{Zn}$ | $c_{Zn}$ | $L$ |
|---|---|---|---|---|
|  | (-) | (µM) | (µM$^{-1}$) | (µM) |
| **reference value** | $6×10^{-3}$ | $3×10^{-5}$ | 0.32 | $1.2×10^{-3}$ |
| **search range** | $6×10^{-4} – 9×10^{-3}$ | $4×10^{-6} – 6×10^{-5}$ | $0.16 – 7.05$ | $2.5×10^{-4} – 3.75×10^{-3}$ |

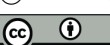



(a)

complexity of uncertainty →

| target field | | | |
|---|---|---|---|
| reference field<br><br>MITgcm-2.8 simulation, annually averaged TMs | reference field<br>+<br>*random noise* $\sim N(0, \varepsilon_{Zn})$ | MITgcm-2.8 simulation, seasonal cycle | MITgcm-ECCO simulation, annually averaged TMs |
| TWIN_ALL | synObs_ALL_noise | synObs_ALL_seas | synObs_ALL_circ |
| | | synObs_IDP+_seas | synObs_IDP+_circ |
| TWIN_IDP | synObs_IDP_noise | synObs_IDP_seas | synObs_IDP_circ |

data reduction ↓

(b)

(c)

(d)

(e)



**Figure 1. (a)** Overview of the experiments carried out in this study. For experiments in the first row (ALL), optimisation was carried out using full data coverage. For experiments in the second and third row, modelled and target fields were interpolated to the 3-D geographical coordinates of locations which have Zn data available in an extended version of the GEOTRACES IDP2017 (IDP+; red dots in panel **b**) and in the original version of this data product (IDP, blue dots in **b**). **(c)** Distribution of IDP2017(+) observations in model depth layers of MITgcm-2.8. **(d)** Taylor diagram comparing target fields of the simulations listed in panel **(a)** to the reference field; **(e)** same as **(d)** but with data coverage limited to IDP2017 coordinates.





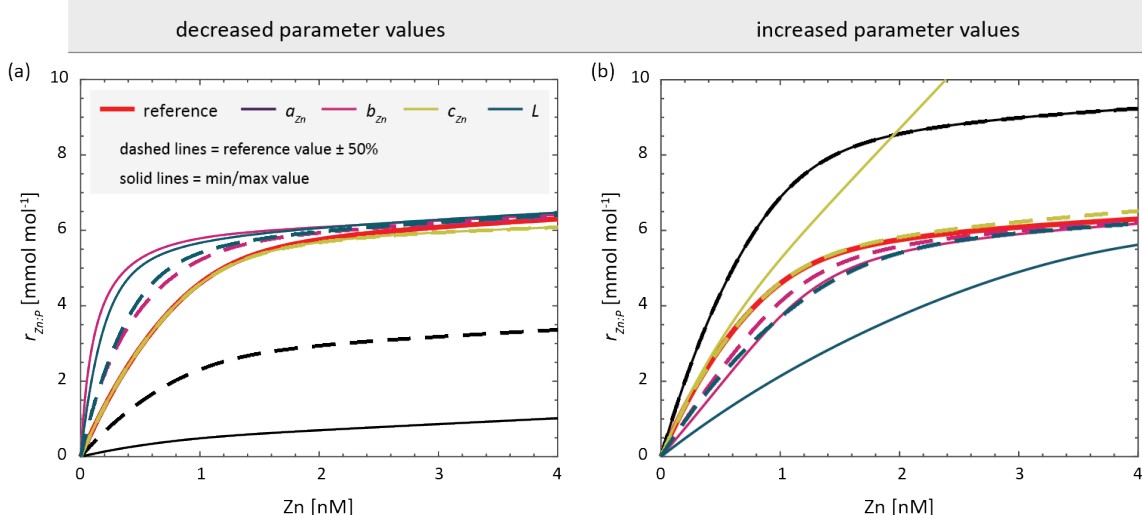

**Figure 2.** Zn:P stochiometry of simulated uptake obtained with reference parameters (red) and by separately varying the parameters of Eq.
(2). The stochiometries are calculated with one parameter being **(a)** decreased or **(b)** increased by 50% (dashed lines) or set to the
boundary value in Table 1 (solid line).





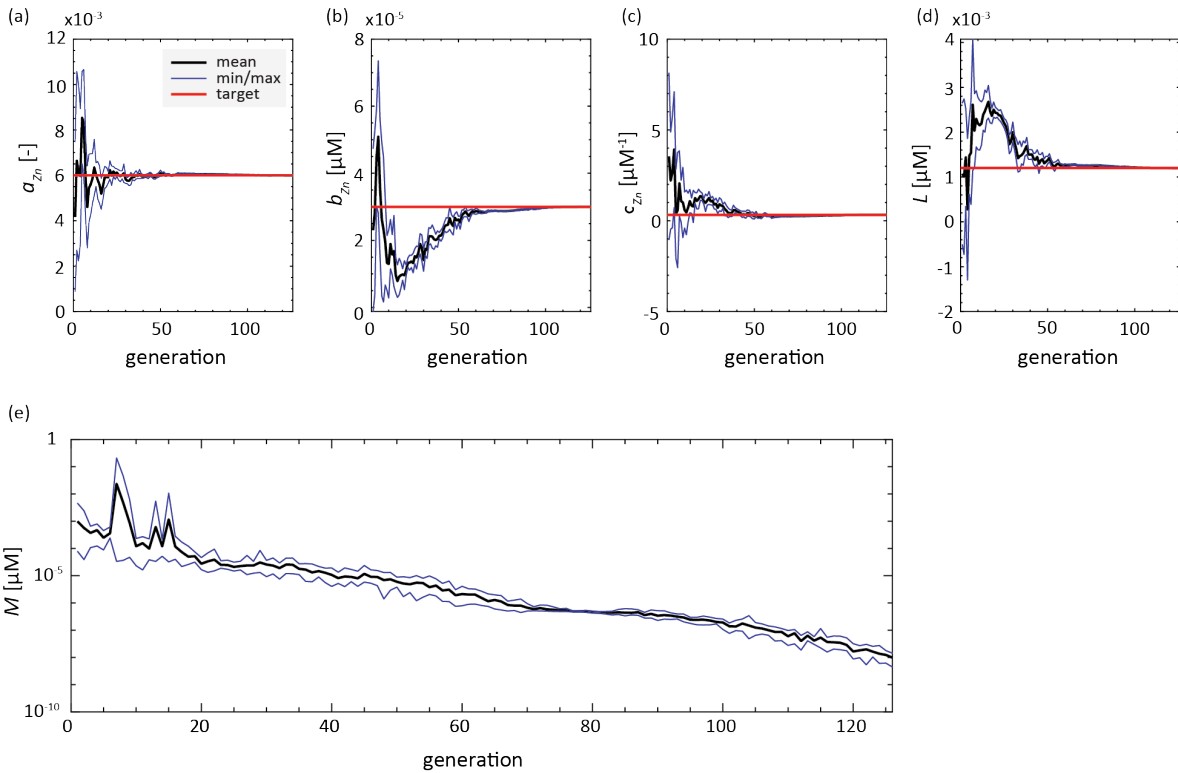

**Figure 3.** Evolution of parameter values and misfit (RMSE) in experiment TWIN_ALL. Red lines indicate the target parameter values. Black trajectories show the mean parameter value over all individuals in each generation of ten individuals, while blue lines mark maximum and minimum parameter values in the generation.

1020





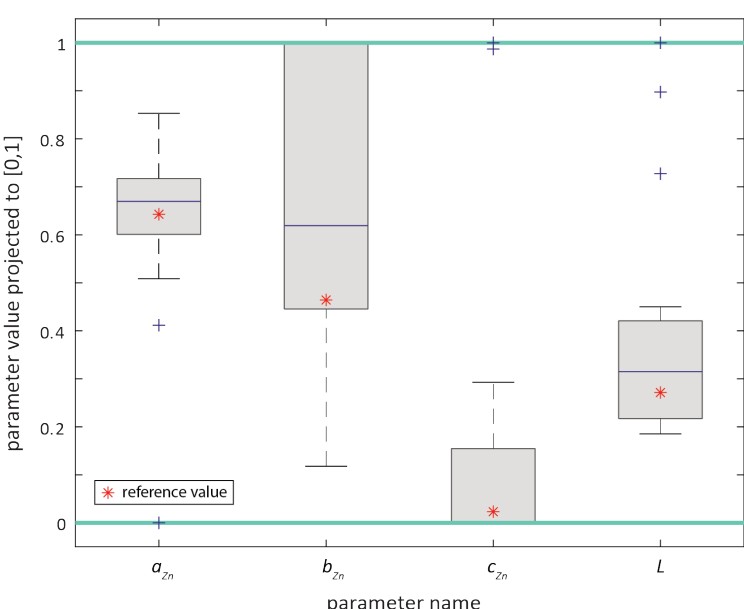

**Figure 4**. Boxplots of optimised parameter values obtained in all synObs_experiments, presented in parameter space rescaled to the interval [0,1], i.e. turquoise lines at 0 and 1 correspond to boundary values for each parameter. Red stars indicate the reference parameter values.





**Figure 5**. Results from the last generation of all synObs experiments. Error bars represent 2 standard deviations calculated using the 10 individuals of the last genaration. Dots represent parameter values resulting in minimum misfit within the prescribed boundaries during optimisation, coloured according to misfit metric.





**Figure 6.** Zn uptake systematics resulting from optimised parameter sets obtained in synObs experiments. Dashed lines in **(e)** and **(f)** are results obtained with the IDP+ data coverage (see text).





**Figure 7**. Maps of surface Zn concentration **(a-c)**, and zonal mean Zn concentration for the Atlantic **(d-f)** and the Pacific **(g-i)**, showing the RMSE-optimised Zn field obtained in synObs_ALL_seas (first column), the difference between this field and the target field (second column), and the difference between this field and the VolRMSE-optimised model Zn field from the same experiment (third column). Note the different colour scales between column 2 and 3.



**Figure 8**. Maps of surface Zn concentration **(a-c)**, and zonal mean Zn concentration for the Atlantic **(d-f)** and the Pacific **(g-i)**, showing the RMSE-optimised Zn field obtained in synObs_ALL_circ (first column), the difference between this field and the target field (second column), and the difference between this field and the VolRMSE-optimised model Zn field from the same experiment (third column). Note 1045 the different colour scales in column 2 and 3. All results are interpolated to the grid of MITgcm-ECCO.





**Figure 9**. Comparison of parameter values obtained in synObs_IDP (empty symbols) or synObs_IDP+ (filled symbols) with those from the corresponding synObs_ALL experiment. Symbols indicate the experiment type.

1050







**Figure 10. (a, c)** Integrated reference residuals (i.e. $Zn_{reference}$ - $Zn_{target}$) and residuals of optimised model over each model depth level. **(f, h)** are zoom-ins of **(a, c)**. **(b, g)** and **(d, i)** are the same as **(a, f)** and **(c, h)** but the integration is done with residuals being squared and with absolute values of residuals respectively. In order to make results obtained with the two data situations comparable, all results are normalised by the total number of observations. Note that positive and negative residuals within the same depth level can cancel each other out, and that the sum is influenced by the number of model cells located in the corresponding model depth level, whereof the distribution is shown in **(e, j)**.





**Figure 11.** Comparison of Zn obtained in synObs_IDP_circ to Zn obtained in synObs_ALL_circ. The first row compares RMAE-optimised Zn obtained in synObs_IDP_circ and synObs_ALL_circ. The second and third row show the effect on deviations seen in the first row if Zn is either squared, as in RMSE, or five ocean regions are distinguished, as in BasinRMAE. The first and second column plot RMAE-optimised Zn obtained in synObs_IDP_circ and synObs_ALL_circ restricted to the coordinates in the GEOTRACES IDP2017v2 and the entire field, respectively. Panels **(e)** and **(h)** of the third column compare Zn obtained with RMSE misfit function and BasinRMAE misfit function in synObs_IDP_circ to the corresponding Zn obtained in synObs_ALL_circ, respectively.