# Peer review of "Influence of GEOTRACES data distribution and misfit function choice on objective parameter retrieval in a marine zinc cycle model"

_Biogeosciences, 2022_

## Author Comment (AC1)

**Answer to Anonymous Referee #1:**

*The authors used numerical models and an optimization technique to explore how the spatially variable Zn:P uptake ratios by phytoplankton can be constrained by synthetic observations. The authors focused on how limited spatial data coverage, model circulation uncertainty, and the choice of objective function, which constitute three major sources of uncertainty in data-model assimilation studies, influence the optimization results. I found the results informative and potentially important to the marine bio-geoscience community. The manuscript is overall well-written except for some minor points listed below.*

We thank Anonymous Referee #1 for taking the time to review our manuscript and for her/his positive assessment and valuable comments.
Below, we include our detailed answers to all comments and questions.

***Main point:***
*(1) The only major concern is that this study might appear to be esoteric and technical to many readers if model-derived synthetic data are the only target. More specifically, the numerical experiments performed and the results discussed here seem to be an important preliminary step towards using the real (not synthetic) observations to constrain the model formulation of the variable Zn:P uptake ratios. Why don't the authors use the best observation data coverage, the best ocean circulation model, and the best objective function to suggest the best estimate for the relationship between Zn:P uptake ratios and Zn? Has this optimization been done already or is this beyond the scope of the current study? When I came to the section 3.7, I expected something along the line, but was disappointed by reading what has been already written and some discussions only. Perhaps, a previous study already found an optimal estimate for the parameter set, which was used as a reference parameter set in this study? Even if so, it would be worth being stated.*
We appreciate the questions raised in this comment, and have re-focused Section 3.7 in response (the revised text is included below).
Model optimisations to real data have indeed been carried out previously, without analysis of the sensitivity to data distribution or misfit function. Our choice to focus on synthetic observations is motivated by the need to assess such sensitivities. In the revised manuscript, we (i) explicitly raise the awareness of the currently largely lacking discussion about impacts of the choice of misfit function and model imperfections, (ii) refer to a previous optimisation study using real Zn data, and (iii) highlight how implications can be drawn from our study with regard to model calibration towards real data *and* with regard to interpretation/discussion of optimisation results. We hope that this addresses the referee's concerns about Section 3.7. Elsewhere in the manuscript, we have extended our reasoning for discussing the results mainly in the light of uptake systematics and large-scale export fluxes when we introduce the Figure that shows the results of synObs experiments as their resulting uptake systematics (Fig. 6), and we clarified in the Methods section that the choice of our reference parameter set was based on Eq. (2) to best fit to *E. huxleyi BT6* culture data of Sunda and Huntsman (1992).
More broadly, we would like to strongly assert that our analysis of the influence of data distribution and subjective choices during optimisation is fundamental to the question of what scientific inferences can be drawn from objective model optimisation. Assessing this influence rigorously is only possible with full control on the observations, hence our study's focus on optimisation to synthetic observations. As such, we would contend that it is neither esoteric nor technical.

Revised Section 3.7 now reads:
"Although there are modelling studies of marine trace metal cycling that objectively calibrate a variety of their model parameters (e.g. Frants et al., 2016; Weber et al., 2018; Pasquier et al., 2022), the impact of data distribution, model

imperfections or choice of misfit function on optimisation results are often not discussed. In this study, we have separately assessed how optimisation results are impacted by these sources of uncertainty. In accordance with other work (e.g. Löptien and Dietze, 2019; Kriest et al., 2020), our ensemble of optimisations shows that biogeochemical parameters are often optimised to compensate for the inability of model formulations to reproduce the target field (Sect. 3.3). Reconstructed Zn uptake systematics were most different from the reference uptake systematics in experiments with systematic differences between the large-scale circulation of the model and that underlying the target field (synObs_circ; Fig. 6); misfits obtained in this experiment type were also about an order of magnitude higher than those for synObs_seas, which differ only in terms of the presence or absence of seasonality within the same physical model. While optimisation to real Zn data (e.g. Weber et al., 2018) is outside this study's focus, the results of our ensemble have direct implications for such optimisations and the inferences that may be drawn from them:

- Because biogeochemical parameters are often optimised to compensate for the inability of model formulations to reproduce the target field, any optimisation of a simple biogeochemical model such as ours towards real data must be seen as attempting to retrieve the systematics of biogeochemical behaviour, rather than physically meaningful parameter values. This is especially the case since – even though observations from wild phytoplankton (Twining and Baines, 2013) indicate geographical systematics that are similar to those that result from this model formulation (de Souza et al., 2018) – there is no reason to believe that the stoichiometry of Zn:P uptake in the real ocean should follow a single dependence on dissolved Zn concentration.
- Because increasing both spatial and temporal model resolution might be computationally unaffordable, even for a relatively efficient global optimisation algorithm like CMA-ES, it is important for studies focusing on optimisation towards global (micro)nutrient distributions with long whole-ocean residence times to prioritise the choice of circulation model, with special focus on accurate simulation of large-scale circulation timescales.
- It is important to recognise the subjectivity that the choice of misfit function introduces to objective parameter optimisation, and to carefully weigh the sensitivities implicit to the misfit function in making this choice for any particular application. A misfit function that appears suitable for optimisation in a simple TWIN experiment, in which the model can perfectly describe the target field, may not be the best choice for optimisation towards noisy, incomplete and/or irregularly distributed real-world data.

It should also be emphasised that our study has not considered the influence of model simplifications, such as the lack of external sources of Zn or simplifications in the underlying P cycling model. External inputs such as those from marginal sediments, atmospheric deposition, or hydrothermal vents (e.g. Conway and John, 2014; Roshan et al., 2016; Lemaitre et al., 2020; Liao et al., 2020) are not relevant to our optimisation ensemble to synthetic observations, but their potential significance should be taken into account during optimisation to real data. With regard to the underlying P cycling model, it directly affects Zn cycling in our model formulation, since Zn uptake is related to $PO_4$ uptake through $r_{Zn:P}$ (Sunda and Huntsman, 1992), and Zn remineralises with the same globally constant length-scale as P (Twining et al., 2014). […]".

***Minor points:***

*(1) I can see the role of three parameters, i.e., a, b, and c in Equation (2), in determining the relationship between Zn:P uptake ratio and Zn. However, how the ligand concentration L is controlling the relationship is not clear. Is there a formulation relating L and the Zn:P uptake ratio or a formulation relating L and Zn2+?*

Thanks for pointing this out (Referee 2 raises a similar point). We have introduced the derivation of $Zn^{2+}$ from our tracer Zn and the ligand concentration L. The revised paragraph in Sect. 2.1.2 now reads:

"Following Ellwood and Van den Berg (2000), concentrations of $Zn^{2+}$ are calculated from total dissolved Zn (the tracer carried in the model) in two steps: first, by assuming rapid equilibration of non-ligand-bound Zn (Zn') with an organic ligand with conditional stability constant $K_L=10^{10}$ $M^{-1}$ and spatially constant concentration, which allows calculation of Zn' by solving the quadratic equation:

$$K_L \cdot (Zn')^2 + (K_L \cdot L - K_L \cdot Zn + 1) \cdot Zn' - Zn = 0 \tag{3}$$

and second, by calculating $Zn^{2+}$ from Zn' using the inorganic side-reaction coefficient $\alpha_{Zn} = 2.1$:

$$Zn^{2+} = \frac{Zn'}{\alpha_{Zn}} \tag{4}"$$

*(2) In Figure 2, the line color for the parameter "a" does not match between the legend and plots. In the legend, it looks like purple to me while it is black in the figures.*

That was a mistake in our Fig.; we have edited the legend.

*(3) I am having difficulty in interpreting Figure 10. What do the X-axes represent in panels (e) and (j)? What do the different lines represent in other panels? (e.g., 'refALL', 'refIDP', 'xALL', and 'xIDP')? Are these labels defined in the text or in the figure caption?*

We agree that Fig. 10 and its legend can be clarified, and have edited it (see below). In panels (e) and (j), we show relative frequency distributions considering only the vertical distribution of observations – this is now explicitly stated by an axis label. We distinguish between 3 different observational sets (MITgcm-2.8deg and IDP17[+]), as clarified in the legend below this. We have also moved the legend for panels a-d and h-i, and hope the figure is more legible as a result.

*(4) The authors discussed additional uncertainties that would rise when applying the optimization to the real observation data in Section 3.7. What about uncertainty in external inputs of Zn (aeolian deposition and coastal sediments, etc.) to the ocean surface? Is it minor compared to the uncertainty associated with model parameterizations of biogenic Zn cycles?*

We agree. While not relevant to our optimisations (since there are no external inputs influencing the synthetic observations) these processes may be relevant for optimisations to real data. We have added a short discussion of this in Section 3.7 (see text of this section above).

[Figure]

Caption: "Figure 10. […] Panel **(e)** shows relative frequency distributions of the vertical distribution of three different observational sets, which are the non-reduced observations, i.e. MITgcm-2.8deg, IDP2017, and IDP2017+ and panel **(j)** represents a zoom-in thereof."

**References**

Conway, T. M. and John, S. G.: The biogeochemical cycling of zinc and zinc isotopes in the North Atlantic Ocean, Global Biogeochemical Cycles, 28, 1111-1128, https://doi.org/10.1002/2014gb004862, 2014.

de Souza, G. F., Khatiwala, S. P., Hain, M. P., Little, S. H., and Vance, D.: On the origin of the marine zinc–silicon correlation, Earth and Planetary Science Letters, 492, 22-34, https://doi.org/10.1016/j.epsl.2018.03.050, 2018.

Ellwood, M. J. and Van den Berg, C. M. G.: Zinc speciation in the Northeastern Atlantic Ocean, Marine chemistry, 68, 295-306, https://doi.org/10.1016/S0304-4203(99)00085-7, 2000.

Frants, M., Holzer, M., DeVries, T., and Matear, R.: Constraints on the global marine iron cycle from a simple inverse model, Journal of Geophysical Research: Biogeosciences, 121, 28-51, https://doi.org/10.1002/2015jg003111, 2016.

Kriest, I., Kähler, P., Koeve, W., Kvale, K., Sauerland, V., and Oschlies, A.: One size fits all? Calibrating an ocean biogeochemistry model for different circulations, Biogeosciences, 17, 3057-3082, https://doi.org/10.5194/bg-17-3057-2020, 2020.

Lemaitre, N., de Souza, G. F., Archer, C., Wang, R.-M., Planquette, H., Sarthou, G., and Vance, D.: Pervasive sources of isotopically light zinc in the North Atlantic Ocean, Earth and Planetary Science Letters, 539, 116216, https://doi.org/10.1016/j.epsl.2020.116216, 2020.

Liao, W. H., Takano, S., Yang, S. C., Huang, K. F., Sohrin, Y., and Ho, T. Y.: Zn Isotope Composition in the Water Column of the Northwestern Pacific Ocean: The Importance of External Sources, Global biogeochemical cycles, 34, n/a, https://doi.org/10.1029/2019GB006379, 2020.

Löptien, U. and Dietze, H.: Reciprocal bias compensation and ensuing uncertainties in model-based climate projections: pelagic biogeochemistry versus ocean mixing, Biogeosciences, 16, 1865-1881, https://doi.org/10.5194/bg-16-1865-2019, 2019.

Pasquier, B., Hines, S. K. V., Liang, H., Wu, Y., Goldstein, S. L., and John, S. G.: GNOM v1.0: an optimized steady-state model of the modern marine neodymium cycle, Geosci. Model Dev., 15, 4625-4656, https://doi.org/10.5194/gmd-15-4625-2022, 2022.

Roshan, S., Wu, J., and Jenkins, W. J.: Long-range transport of hydrothermal dissolved Zn in the tropical South Pacific, Marine chemistry, 183, 25-32, https://doi.org/10.1016/j.marchem.2016.05.005, 2016.

Sunda, W. G. and Huntsman, S. A.: Feedback interactions between zinc and phytoplankton in seawater, Limnology and Oceanography, 37, 25-40, https://doi.org/10.4319/lo.1992.37.1.0025, 1992.

Twining, B. S. and Baines, S. B.: The Trace Metal Composition of Marine Phytoplankton, Annual review of marine science, 5, 191-215, https://doi.org/10.1146/annurev-marine-121211-172322, 2013.

Twining, B. S., Nodder, S. D., King, A. L., Hutchins, D. A., LeCleir, G. R., DeBruyn, J. M., Maas, E. W., Vogt, S., Wilhelm, S. W., and Boyd, P. W.: Differential remineralization of major and trace elements in sinking diatoms, Limnology and oceanography, 59, 689-704, https://doi.org/10.4319/lo.2014.59.3.0689, 2014.

Weber, T., John, S., Tagliabue, A., and DeVries, T.: Biological uptake and reversible scavenging of zinc in the global ocean, Science (American Association for the Advancement of Science), 361, 72-76, https://doi.org/10.1126/science.aap8532, 2018.

---

## Author Comment (AC2)

**Answer to Anonymous Referee #2:**

We thank Anonymous Referee #2 for her/his comments and suggestions.
Below, we include our detailed answers to all comments and questions.

*The manuscript investigates global spatial distribution of Zn:P uptake ratios by phytoplankton using the GEOTRACES dataset, MIT numerical model and optimization techniques. The methodology is adapted and well described, and analyses focuses on the influence of limited spatial data coverage, model circulation uncertainty, and choices of optimization functions. The manuscript provides some information of interest for the marine biology community. However the manuscript is very dense and technical, and in its actual shape not very accessible for the majority of scientists. I recommend to rework on the manuscript to make it more accessible, to highlight more the conclusions of interest for the zinc cycle and to base the discussions less on the technical aspects. in particular, focus more on the impact of the partial coverage of the database, the seasonal cycle and the circulation, and put more indented the different optimization techniques and synthesize may be in an appendix how their choice can strongly influence the main conclusions of the paper Zn cycle.*

We are glad that the referee finds our manuscript of interest for the community. Regarding accessibility and density, we appreciate the referee's comments, but would like to point out that:

(a) as we state in the original manuscript, the aim of our study is to assess the sensitivities that need to be considered during model optimisation to (GEOTRACES) data, and this indeed motivates the design of our study. As a consequence, the aspects relating to optimisation sensitivity *must* be the focus of our discussion, rather than the zinc cycle more generally. This also represents the novel aspect of this work, which has not been considered by previous Zn model optimisations (see response to Referee 1).

(b) while we would agree that the manuscript is long, we have carefully attempted to structure it such that its sections can be read independently of each other. Thus, we hope that the manuscript structure makes it possible for readers to find the relevant information easily, e.g. reading only Sect. 3.5 if interested in the influence of misfit metric on optimisation.

Nonetheless, in response to the referee's comment, we have made revisions throughout the manuscript with the aim of broader accessibility. We note that the journal's guidelines do not allow the supplementary material to contain "scientific interpretations or findings that would go beyond the contents of the manuscript", but have tried to improve accessibility of the main text, such as:

- by providing synthesis sentences that summarise the main findings discussed in a section (e.g. L392-393, L607-608 of the revised manuscript)
- by expanding the reasoning why we discuss results in the context of uptake systematics and export flux. The introduction of Fig. 6 in the beginning of Section 3.2 now reads:
  "Figure 6 illustrates […]. We use this emergent relationship as a measure of model similarity since it controls the geographical systematics of Zn uptake and export (de Souza et al., 2018), although of course the stoichiometry of Zn:P uptake in the *real* ocean is not likely to follow a single dependence on Zn concentration."
- By changing Section 3.7 in order to manifest the importance of our study and its position at the theory-practice interface, and by listing the main implications that can be drawn from our study with regard to model calibration towards real data *and* with regard to interpretation/discussion of optimisation results (see response to Referee 1).

Revised Section 3.7 now reads:

"Although there are modelling studies of marine trace metal cycling that objectively calibrate a variety of their model parameters (e.g. Frants et al., 2016; Weber et al., 2018; Pasquier et al., 2022), the impact of data distribution, model imperfections or choice of misfit function on optimisation results are often not discussed. In this study, we have separately assessed how optimisation results are impacted by these sources of uncertainty. In accordance with other work (e.g. Löptien and Dietze, 2019; Kriest et al., 2020), our ensemble of optimisations shows that biogeochemical parameters are often optimised to compensate for the inability of model formulations to reproduce the target field (Sect. 3.3). Reconstructed Zn uptake systematics were most different from the reference uptake systematics in experiments with systematic differences between the large-scale circulation of the model and that underlying the target field (synObs_circ; Fig. 6); misfits obtained in this experiment type were also about an order of magnitude higher than those for synObs_seas, which differ only in terms of the presence or absence of seasonality within the same physical model. While optimisation to real Zn data (e.g. Weber et al., 2018) is outside this study's focus, the results of our ensemble have direct implications for such optimisations and the inferences that may be drawn from them:

- Because biogeochemical parameters are often optimised to compensate for the inability of model formulations to reproduce the target field, any optimisation of a simple biogeochemical model such as ours towards real data must be seen as attempting to retrieve the systematics of biogeochemical behaviour, rather than physically meaningful parameter values. This is especially the case since – even though observations from wild phytoplankton (Twining and Baines, 2013) indicate geographical systematics that are similar to those that result from this model formulation (de Souza et al., 2018) – there is no reason to believe that the stoichiometry of Zn:P uptake in the real ocean should follow a single dependence on dissolved Zn concentration.

- Because increasing both spatial and temporal model resolution might be computationally unaffordable, even for a relatively efficient global optimisation algorithm like CMA-ES, it is important for studies focusing on optimisation towards global (micro)nutrient distributions with long whole-ocean residence times to prioritise the choice of circulation model, with special focus on accurate simulation of large-scale circulation timescales.

- It is important to recognise the subjectivity that the choice of misfit function introduces to objective parameter optimisation, and to carefully weigh the sensitivities implicit to the misfit function in making this choice for any particular application. A misfit function that appears suitable for optimisation in a simple TWIN experiment, in which the model can perfectly describe the target field, may not be the best choice for optimisation towards noisy, incomplete and/or irregularly distributed real-world data.

It should also be emphasised that our study has not considered the influence of model simplifications, such as the lack of external sources of Zn or simplifications in the underlying P cycling model. External inputs such as those from marginal sediments, atmospheric deposition, or hydrothermal vents (e.g. Conway and John, 2014; Roshan et al., 2016; Lemaitre et al., 2020; Liao et al., 2020) are not relevant to our optimisation ensemble to synthetic observations, but their potential significance should be taken into account during optimisation to real data. With regard to the underlying P cycling model, it directly affects Zn cycling in our model formulation, since Zn uptake is related to $PO_4$ uptake through $r_{Zn:P}$ (Sunda and Huntsman, 1992), and Zn remineralises with the same globally constant length-scale as P (Twining et al., 2014). […]".

*Also, the study on the influence of the circulation needs to be better described and analyzed. While for the other analyzes there is information on the range of variation of the parameters or the techniques, no details are given on the sensitivity experiment with the higher resolution model. the paper concludes on the strong influence of the ocean circulation, but perhaps the circulations generated by the models are very different, and only caused by significant shortcomings in the low-resolution model, so that range of variation is not realistic.*

We are not entirely sure what the referee means here. Firstly, our manuscript does not particularly emphasise or "[conclude] on the strong influence of the ocean circulation". This is simply one aspect that we discuss, and indeed we emphasise the importance of misfit function choice more strongly – including in the title, abstract and conclusions of the original manuscript. Secondly, we do not quite understand what the referee means regarding "information on range of variation and parameters": in Section 2.3.1, we describe the setup of our circulation experiments to the same (or greater) level of detail as the other experiments.

More broadly, however, we agree that the circulations generated by MITgcm-2.8 and MITgcm-ECCO are quite different, especially regarding the timescale of deep-ocean ventilation, as discussed in Section 3.3.3 of the original manuscript. As noted in Section 2.3.1 of the original manuscript, it would be beyond the scope of this study to identify all differences between the two MITgcm versions that finally cause the differences in the retrieved parameters. We have tried to summarise the most relevant characteristics (spatial differences in Zn concentrations and export fluxes (Sect. 2.3.1) as well as ideal age distributions (Sect. 3.3.3)).

Additionally, we agree that our finding regarding the sensitivity of different circulation models to misfit functions could be more or less pronounced, depending on the circulation model and reference values that were chosen. However, we remain confident that the VolRMSE misfit metric keeps being more prone to compensate for large-scale differences in circulation patterns, in particular patterns related to deep ocean ventilation. To clarify this, in the revised manuscript we have introduced a note that the degree of variation in uptake systematics may depend on differences in the underlying circulations and the choice of reference parameter in Section 3.3.3, where differences in the underlying circulation are discussed

***Minor points :***
*- Parameterization of Zn2+/Zn with Ligands (L) is not described*
Thanks - we have introduced the derivation of $Zn^{2+}$ from our tracer Zn and the ligand concentration L. The revised paragraph in Sect. 2.1.2 now reads:
"Following Ellwood and Van den Berg (2000), concentrations of $Zn^{2+}$ are calculated from total dissolved Zn (the tracer carried in the model) in two steps: first, by assuming rapid equilibration of non-ligand-bound Zn (Zn') with an organic ligand with conditional stability constant $K_L=10^{10}$ M$^{-1}$ and spatially constant concentration, which allows calculation of Zn' by solving the quadratic equation:

$$K_L \cdot (Zn')^2 + (K_L \cdot L - K_L \cdot Zn + 1) \cdot Zn' - Zn = 0 \qquad (3)$$

and second, by calculating $Zn^{2+}$ from Zn' using the inorganic side-reaction coefficient $\alpha_{Zn} = 2.1$:

$$Zn^{2+} = \frac{Zn'}{\alpha_{Zn}} \qquad (4)"$$

*- the czn parameter seems to be almost systematically optimized by the lowest allowed value. wouldn't it be possible to widen the to allow lower values?*
It is correct that the parameter $c_{Zn}$ converged to its lower boundary in almost all optimisations. We tested the effect of reducing the lower boundary for $c_{Zn}$ to the physically meaningless value of -0.16 µM$^{-1}$ for two RMSE-optimisations: synObs_IDP_noise, where $c_{Zn}$ was the only parameter that converged to its lower boundary value, and

synObs_ALL_circ, where $b_{Zn}$ as well as $c_{Zn}$ converged to boundary values. In both experiments, the optimum value of $c_{Zn}$ was found at the new lower bound of -0.16 $\mu M^{-1}$, while the other parameters changed by a maximum of 25 % relative to the corresponding previous optimisations. While in one experiment the minimum misfit found by CMA-ES was lower (-0.052 %) than the misfit identified with default parameter boundaries, in the other, widening the lower boundary of parameter $c_{Zn}$ resulted in a misfit that was minimally higher (+0.006%) than the final misfit reached with default boundaries. Thus, expanding the lower boundary for $c_{Zn}$ led CMA-ES to terminate in a local misfit minimum in this experiment.

In our original manuscript, we mentioned that we carried out experiments with widened boundaries [L. 305-310]. We decided not to show the detailed results mentioned above as they are not new but rather confirm findings that have already been shown in previous studies (Kriest et al., 2017; Falls et al., 2021), and we prefer to keep that as is.

**References**

Conway, T. M. and John, S. G.: The biogeochemical cycling of zinc and zinc isotopes in the North Atlantic Ocean, Global Biogeochemical Cycles, 28, 1111-1128, https://doi.org/10.1002/2014gb004862, 2014.

de Souza, G. F., Khatiwala, S. P., Hain, M. P., Little, S. H., and Vance, D.: On the origin of the marine zinc–silicon correlation, Earth and Planetary Science Letters, 492, 22-34, https://doi.org/10.1016/j.epsl.2018.03.050, 2018.

Ellwood, M. J. and Van den Berg, C. M. G.: Zinc speciation in the Northeastern Atlantic Ocean, Marine chemistry, 68, 295-306, https://doi.org/10.1016/S0304-4203(99)00085-7, 2000.

Falls, M., Bernardello, R., Castrillo, M., Acosta, M., Llort, J., and Galí, M.: Use of Genetic Algorithms for Ocean Model Parameter Optimisation, Geosci. Model Dev. Discuss., 2021, 1-44, https://doi.org/10.5194/gmd-2021-222, 2021.

Frants, M., Holzer, M., DeVries, T., and Matear, R.: Constraints on the global marine iron cycle from a simple inverse model, Journal of Geophysical Research: Biogeosciences, 121, 28-51, https://doi.org/10.1002/2015jg003111, 2016.

Kriest, I., Sauerland, V., Khatiwala, S., Srivastav, A., and Oschlies, A.: Calibrating a global three-dimensional biogeochemical ocean model (MOPS-1.0), Geoscientific model development, 10, 127-154, https://doi.org/10.5194/gmd-10-127-2017, 2017.

Kriest, I., Kähler, P., Koeve, W., Kvale, K., Sauerland, V., and Oschlies, A.: One size fits all? Calibrating an ocean biogeochemistry model for different circulations, Biogeosciences, 17, 3057-3082, https://doi.org/10.5194/bg-17-3057-2020, 2020.

Lemaitre, N., de Souza, G. F., Archer, C., Wang, R.-M., Planquette, H., Sarthou, G., and Vance, D.: Pervasive sources of isotopically light zinc in the North Atlantic Ocean, Earth and Planetary Science Letters, 539, 116216, https://doi.org/10.1016/j.epsl.2020.116216, 2020.

Liao, W. H., Takano, S., Yang, S. C., Huang, K. F., Sohrin, Y., and Ho, T. Y.: Zn Isotope Composition in the Water Column of the Northwestern Pacific Ocean: The Importance of External Sources, Global biogeochemical cycles, 34, n/a, https://doi.org/10.1029/2019GB006379, 2020.

Löptien, U. and Dietze, H.: Reciprocal bias compensation and ensuing uncertainties in model-based climate projections: pelagic biogeochemistry versus ocean mixing, Biogeosciences, 16, 1865-1881, https://doi.org/10.5194/bg-16-1865-2019, 2019.

Pasquier, B., Hines, S. K. V., Liang, H., Wu, Y., Goldstein, S. L., and John, S. G.: GNOM v1.0: an optimized steady-state model of the modern marine neodymium cycle, Geosci. Model Dev., 15, 4625-4656, https://doi.org/10.5194/gmd-15-4625-2022, 2022.

Roshan, S., Wu, J., and Jenkins, W. J.: Long-range transport of hydrothermal dissolved Zn in the tropical South Pacific, Marine chemistry, 183, 25-32, https://doi.org/10.1016/j.marchem.2016.05.005, 2016.

Sunda, W. G. and Huntsman, S. A.: Feedback interactions between zinc and phytoplankton in seawater, Limnology and Oceanography, 37, 25-40, https://doi.org/10.4319/lo.1992.37.1.0025, 1992.

Twining, B. S. and Baines, S. B.: The Trace Metal Composition of Marine Phytoplankton, Annual review of marine science, 5, 191-215, https://doi.org/10.1146/annurev-marine-121211-172322, 2013.

Twining, B. S., Nodder, S. D., King, A. L., Hutchins, D. A., LeCleir, G. R., DeBruyn, J. M., Maas, E. W., Vogt, S., Wilhelm, S. W., and Boyd, P. W.: Differential remineralization of major and trace elements in sinking diatoms, Limnology and oceanography, 59, 689-704, https://doi.org/10.4319/lo.2014.59.3.0689, 2014.

Weber, T., John, S., Tagliabue, A., and DeVries, T.: Biological uptake and reversible scavenging of zinc in the global ocean, Science (American Association for the Advancement of Science), 361, 72-76, https://doi.org/10.1126/science.aap8532, 2018.